# Bridging Neural and Symbolic Representations with Transitional Dictionary Learning

**Junyan Cheng**
Thayer School of Engineering
Dartmouth College
Hanover, NH 03755, USA
`jc.th@dartmouth.edu`

**Peter Chin**
Thayer School of Engineering
Dartmouth College
Hanover, NH 03755, USA
`pc@dartmouth.edu`

## Abstract

This paper introduces a novel **Transitional Dictionary Learning** (TDL) framework that can implicitly learn symbolic knowledge, such as visual parts and relations, by reconstructing the input as a combination of parts with implicit relations. We propose a game-theoretic diffusion model to decompose the input into visual parts using the dictionaries learned by the Expectation Maximization (EM) algorithm, implemented as the online prototype clustering, based on the decomposition results. Additionally, two metrics, clustering information gain, and heuristic shape score are proposed to evaluate the model. Experiments are conducted on three abstract compositional visual object datasets, which require the model to utilize the compositionality of data instead of simply exploiting visual features. Then, three tasks on symbol grounding to predefined classes of parts and relations, as well as transfer learning to unseen classes, followed by a human evaluation, were carried out on these datasets. The results show that the proposed method discovers compositional patterns, which significantly outperforms the state-of-the-art unsupervised part segmentation methods that rely on visual features from pre-trained backbones. Furthermore, the proposed metrics are consistent with human evaluations.

## 1 Introduction

In recent years, there has been a desire to incorporate the interpretability, compositionality, and logistics of symbolic systems into neural networks (NNs). Existing methods combine them in an ad hoc manner, Dong et al. (2019); Wang et al. (2019) converts symbolic programs into differentiable forms, Cornelio et al. (2023); Segler et al. (2018) introduce symbolic modules to assist NNs, Gupta & Kembhavi (2023) use neural and symbolic modules as building blocks for visual programs. These methods do not truly bring symbolic power to NNs, but simply allow them to work synergically. The essential disparity lies at a low level. NNs use distributed representations, while symbolic systems use symbolic representations. This motivates us to explore ways to bridge neural and symbolic representations, thus combining the two types of intelligence from the ground up.

Cognitive science studies (Taniguchi et al., 2018) have suggested that symbolic representation in the human brain does not appear out of nowhere; rather, there is a gradual transition from neural perception to *preliminary symbols* and eventually to symbolic languages over the course of human evolution, as people observe and interact with their environment. The transitional representation, preliminary symbols, is essential in connecting neural and symbolic representation. Taking this concept into account, we attempt to replicate the process of transitional representation arising from neural representation, through unsupervised learning on visual observations in this study, as an exploration of the potential path of unifying neural and symbolic thinking at the representation level.

Representation explains how the input is made up of reusable components (Bengio et al., 2013). Taking visual observations as an example, distributed representation is explained by vectors, such as principal components, that illustrate the high-dimensional statistical features, and symbolic representation uses structural methods, such as logic sentences, that explain the visual parts and their connections. A transitional representation should be in between that (1) contains high-dimensional details of the input and (2) implies structural information about the semantics of the input.

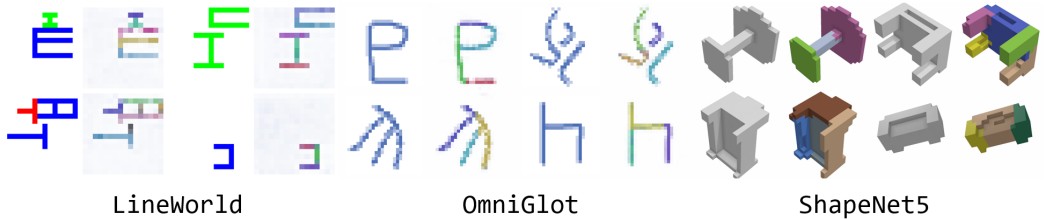

LineWorld    OmniGlot    ShapeNet5

Figure 1: Decomposing samples from three datasets into visual parts marked with different colors, shallowness represents confidence. Odd columns are input, even columns are decompositions.

We propose a novel **Transitional Dictionary Learning (TDL)** framework that implicitly learns symbolic knowledge, such as visual parts and relations, by reconstructing the input as a combination of parts with implicit relations. With a simple fine-tuning that aligns the output with human preference through reinforcement learning and a heuristic reward, the model can give a human-interpretable decomposition of the input; examples are shown in Figure 1. TDL uses an Expectation Maximization (EM) algorithm to iteratively update dictionaries that store hidden representations of symbolic knowledge, through an online prototype clustering on the visual parts decomposed from the inputs by a novel game-theoretic diffusion model using the current dictionaries.

We suggest two metrics to evaluate the learned representation. Clustering Information Gain, which assesses if the learned dictionary is parsimonious and representative, and a heuristic shape score that assesses if the decompositions are in line with human intuition. We conduct unsupervised learning experiments on three abstract compositional visual object datasets, which require the model to utilize the compositionality of data instead of simply visual features, and three tasks on symbol grounding to predefined classes of parts and relations, and transfer learning to unseen classes. The results show huge improvements compared to the state-of-the-art part segmentation baselines, which struggle to process abstract objects that lack distinct visual features. We also conduct human evaluations; the results demonstrate significantly improved interpretability of the proposed method and the proposed metrics are consistent with human assessments. Our contributions are concluded as follows.

- We propose the unsupervised Transitional Dictionary Learning to learn symbolic features in representations with a novel game-theoretic diffusion model and online prototype clustering.
- We introduce two metrics, the clustering information gain and the heuristic shape score, to evaluate the learned representation and give evaluation results agreed to human judgment.
- We perform experiments, compare our method with state-of-the-art unsupervised part segmentation models, and conduct human evaluations for all models and proposed metrics.

Our code and data are available at `https://github.com/chengjunyan1/TDL`.

## 2 RELATED WORK

**Neural-Symbolic Learning.** Some approaches incorporate a symbolic program to assist NNs. Segler et al. (2018) used a tree search in learning retrosynthetic routes, Amizadeh et al. (2020) applied first-order logic to answer visual questions, and Young et al. (2019) introduced a symbolic controller to generate repeating visual patterns. However, they are usually tailored to specific tasks. Program synthesis could be more flexible. Inala et al. (2020) synthesized multi-agent communication policies, Sun et al. (2021) searched for autonomous vehicle control programs, and Gupta & Kembhavi (2023) synthesized visual task scripts. Nevertheless, symbolic thinking was not truly incorporated. Thus, differentiable symbolic modules are proposed. Wang et al. (2019) relaxed an SAT solver as a NN layer, Riegel et al. (2020) made first-order logic propositions differentiable, Dong et al. (2019) used NNs as trainable logical functions, Dai et al. (2019) optimized NN with abductive logic, and Goyal et al. (2021) learned neural production systems of visual entities. However, they overlooked the fact that the disparity between neural and symbolic representations is the source of the problem.

**Compositional Representation.** Fei-Fei & Perona (2005); Csurka et al. (2004) show early efforts to learn compositionality through the bag of words. Hinton (2021) proposed an architecture to learn

part-whole hierarchies, which Garau et al. (2022) implemented using an attention model. Du et al. (2020) employed Energy-Based Models (EBMs) to learn compositional parts for image generation. Chen et al. (2020) learned to compose programs from basic blocks. Mendez et al. (2022) learned reusable compositions for lifelong learning. Lake et al. (2015) used Bayesian Program Learning to compose handwritten characters from parts and strokes. Shanahan et al. (2020) used attention to learn relations among objects. Mao et al. (2019) learned visual concept embeddings and Cao et al. (2021) learned them as prototypes. Kipf et al. (2020) used contrastive learning to embed concepts and implicit relations, and Wu et al. (2022) employed EBMs to embed concepts and relations for zero-shot inference. Du et al. (2021) also used EBM to represent concepts. LeCun (2022) introduced an EBM-based framework to learn hierarchical planning. In comparison, we aim to learn the seamless transition between neural and symbolic representations unsupervisedly.

**Unsupervised Segmentation.** Segmentation extracts structural information from visual inputs. Lin et al. (2020) used spatial attention to learn scene parsing, Bear et al. (2020) utilized physical properties, Kim et al. (2020) clustered on feature maps, and Van Gansbeke et al. (2021) improved it by contrastive learning. Lou et al. (2022) parsed scene graphs by aligning the image with the dependency graph of a caption. Melas-Kyriazi et al. (2022) used deep spectral methods to segment. The parsed elements in these methods are not reusable; co-part segmentation attempts to address it. Collins et al. (2018) learned visual concepts by NMF, Hung et al. (2019); Choudhury et al. (2021) learned reusable parts by self-supervised learning, Amir et al. (2021) extracted concepts from a pre-trained vision Transformer, Gao et al. (2021) learned co-parts utilizing motions in videos, Ziegler & Asano (2022) uncovered parts with a Sinkhorn-Knopp clustering, Yu et al. (2022) utilized capsule networks to discover face parts and He et al. (2022) learned parts by hierarchical image generation. However, they rely on concrete visual features and learn visual patterns instead of discovering compositionality like us, thus not working on abstract input, as shown in experiments in Section 5.

## 3 TRANSITIONAL DICTIONARY LEARNING

We first introduce the transitional representation and its optimization target in Section 3.1, then optimize this target with our TDL framework based on an EM algorithm in Section 3.2. Finally, we propose the clustering information gain to evaluate the learned representation in Section 3.3. We use this convention if it is not specified separately: superscript $\cdot^i$ denotes $i$-arity, superscript $\cdot^{(i)}$ with brackets denotes $i$-th sample in a dataset, and subscript $\cdot_i$ denotes $i$-th visual part in a sample.

### 3.1 TRANSITIONAL REPRESENTATION

Given a visual input $x \in \mathbb{R}^{H \times W \times C}$, suppose 2D here for simplicity without loss of generality, we can compress it into a low-dimensional embedding $r = f(x) \in \mathbb{R}^d$ using a NN or other machine learning models $f$ that minimizes the *reconstruction error* by $\min_r \epsilon(g(r), x)$ with a decoder $g$. As discussed above, such representations lack interpretability, compositionality, and structural information.

Alternatively, we can employ a symbolic representation that explicitly identifies structural information. Predicate logic, the dominant and theoretically *complete* (Newell & Simon, 1976) symbolic representation, expresses the input $x$ as a conjunction of logical statements $\Omega = \rho_1^1(\cdot) \wedge \rho_2^1(\cdot) \wedge ... \wedge \rho_1^2(\cdot, \cdot) \wedge \rho_2^2(\cdot, \cdot) \wedge ...$ that minimizes *semantic distance* $\min_\Omega d_S(\Omega, x)$, where $\rho_i^k$ is the $i$-th logical sentence using a predicate of arity $k$, the number of arguments. Arguments $\cdot$ can be logic variables, constants, or even logical sentences that form high-order sentences.

To simplify the analysis while keeping generality, we assume that an input $x$ is *linearly* composed of visual parts $x_i$ by $x = \sum_{i=1}^{N_P} x_i$, where $N_P$ is the number of parts, although non-linear assumptions exist, such as viewing an image as a stack of layers or projection from a 3D space. To create a *meaningful* logical representation that is semantically close to $x$, we begin with the *entity mappings* $x_i \rightarrow \rho_j^1$, grounding a 1-ary predicate $\rho_j^1 \in D^1$ of entities from the dictionary $D^1$, such as $Cat(x_i)$, $Tree(x_i)$, $Person(x_i)$, in $x_i$. Then, we construct *relation mappings* with higher-ary predicates, taking 2-ary as an example, $(x_i, x_j) \rightarrow \rho_k^2$, where $\rho_k^2 \in D^2$, such as $left\_of(x_i, x_j), larger(x_i, x_j)$. Finally, we depict the *attributes* by predicates such as $Red(x_i), Length(x_i, 5cm)$.

This process is an optimization problem $\arg\max_\Omega P(\Omega|x, D)$ where $D = \{D^1, D^2, ...\}$ is a dictionary collection of different ary predicates. There are two major drawbacks, which also led to

the downfall of symbolic AI. Firstly, symbol grounding that links predicates to visual components is non-trivial. Although it can be automated by supervised learning, annotation is expensive and inflexible. Secondly, designing attribute predicates that capture all the details is impossible.

Therefore, we propose a **transitional representation**, the *neural logic variables* $R = \{r_1, r_2, ..., r_{N_P}\}$ generated by a model $f(x; \theta)$ with a *hidden* dictionary parameterized by $\theta$ as $D$. $R$ is composed of entity vectors $r_i \in \mathbb{R}^d$ and follows the optimization target.

$$\min_\theta \sum_i^N \epsilon(g(R^{(i)}; \theta), x^{(i)}) + \alpha E_{\tilde{D}}(d_S(g_{\tilde{D}}(R^{(i)}; \theta), x^{(i)})) \ where \ R^{(i)} = f(x^{(i)}, \theta) \quad (1)$$

where $R^{(i)} = \{r_j^{(i)}\}_{j=1}^{N_P}$ are variables for $x^{(i)}$ in dataset $X = \{x^{(i)}\}_{i=1}^N$, we omit the parameters of decomposition model $f$ and $g$ that also need to be optimized in this target for simplicity. It minimizes both the reconstruction error of the first "neural" term, where $g(R^{(i)}; \theta) = \sum_{j=1}^{N_P} \hat{g}(r_j^{(i)}; \theta)$ and $\hat{g}(r_j^{(i)}; \theta)$ is the decoder, and the expected semantic distance of all *meaningful* concrete dictionaries by the second "symbolic" term. The predicate head $g_{\tilde{D}}$ can map $R^{(i)}$ to logic sentences $\Omega^{(i)} = g_{\tilde{D}}(R^{(i)})$ that maximally preserve the semantics of the input given a concrete dictionary $\tilde{D}$. $d_S$ is an *ideal* metric that can accurately measure whether two representations express the same semantics. The expectation considers all meaningful dictionaries of an input (e.g., different fonts for the same character), while non-meaningful ones are not considered (e.g., the inputs are dogs, the dictionary is for cats). The coefficient $\alpha$ adjusts the two goals. In practice, we can train an "average" dictionary with transitional representations that have minimal possible distances from all concrete ones, and then align each by fine-tuning. Transitional representation tackles the second problem above by compressing attributes in embeddings and the first problem with unsupervised learning will be discussed in Section 3.2.

## 3.2 EXPECTATION-MAXIMIZATION FOR TRANSITIONAL REPRESENTATION

The first term in Equation 1 can be optimized by the following target (Kreutz-Delgado et al., 2003)

$$\arg\min_\theta \sum_{i=1}^N \epsilon(x^{(i)}, \sum_j^{N_P} \hat{g}(r_j^{(i)}; \theta)) + \lambda \sum_j^{N_P} |r_j^i| \quad (2)$$

optimizes the hidden dictionary $\theta$ by minimizing the reconstruction error from visual parts. However, the key challenge comes from the second term. We consider $R = \{r_i\}_{i=1}^{N_P}$, or its corresponding visual parts, as a *bag of words* for the image $x$ that implicates *hidden* logical sentences $\Omega_R = g_\theta(R)$, where the optimal $\theta^* = \arg\min_\theta E_{\tilde{D}}[d_S(g_{\tilde{D}}(R), g_\theta(R))]$. As we only need to consider meaningful dictionaries $\tilde{D} = \arg\min_{\tilde{D}} d_S(g_{\tilde{D}}(R), x)$ that allows semantically equivalent representations of input $x$, an alternative target for Equation 1 is: $\min_\theta \sum_i^N \epsilon(g(R^{(i)}; \theta), x^{(i)}) + \alpha d_S(x^{(i)}, g_\theta(R^{(i)}))$.

We can reasonably assume that $d_S(x^{(i)}, g_\theta(R^{(i)})) \propto -P(\Omega_{R^{(i)}}|x^{(i)}, \theta)$ where meaningful logic variables and relations are more likely to appear in the dataset than non-meaningful ones, i.e., *reusable* and *compositional*. In other words, the optimal dictionary $\theta^*$ maximizes the likelihood of the dataset. By regarding $x^{(i)}$ as a visual sentence composed of visual words $R^{(i)}$ and dataset $X$ as a visual corpus, we optimize the second term in the alternative target via EM algorithm inspired by the Unigram Language Model (ULM) (Kudo, 2018) which maximizes the likelihood of the dataset by iteratively updating the dictionaries given decomposed visual parts using current dictionary

$$\arg\max_\theta \mathcal{L} = \sum_{i=1}^N \log P(\Omega_{R^{(i)}}|x^{(i)}, \theta) \quad (3)$$

The likelihood of the dataset is computed as the summation of the log-likelihood of the logic representations of all sample $x^{(i)}$ from 1 to $N_A$ arities by $\mathcal{L} = \sum_{i=1}^N \sum_{j=1}^{N_A} \log P(\Omega_{R^{(i)}}^j|x^{(i)}, \theta)$. For 1-ary, we follow ULM by $\log P(\Omega_{R^{(i)}}^1|x^{(i)}, \theta) = \sum_{k=1}^{N_P} \log P(r_k^{(i)})$, for 2-ary, the Markov assumption for sequential data is not suitable, thus we use a joint probability $\log P(\Omega_{R^{(i)}}^2|x^{(i)}, \theta) = \sum_{p=1}^{N_P} \sum_{q=1}^{N_P} \log P(r_p^{(i)}, r_q^{(i)})$, and the same applies for higher arities.

The optimization targets in Equations 2 and 3 give our **Transitional Dictionary Learning (TDL)** framework. Equation 3 can be optimized by clustering all decomposed visual parts, pairs of parts, etc. The complexity increases exponentially with the arity which is unacceptable despite low-ary is adequate to provide a graph-level representation power, we use techniques like online clustering and random sampling to improve the efficiency that are discussed later in Section 4.2. Further discussion of the limitations and the broader impacts of the TDL framework can be found in Appendix L.

### 3.3 Clustering Information Gain

We wish that the learned predicates are *reusable* and *compositional*, the decomposed parts and pairs of the test set should be clustered in as few centroids $\mathcal{C}$ as possible. Thus, we propose Clustering Information Gain (CIG) by comparing the Mean Clustering Error (MCE) of the decomposed parts in the test set, marked $MCE = [\sum_{i=1}^{N} \sum_{j=1}^{N_P} (\min_{c \in \mathcal{C}} ||r_j^{(i)} - c||_2)/N_P]/N$ with the random decomposition $MCE_{rand}$, which is a lower bound when the decomposed terms are randomly scattered, while the best case of MCE is 0 when the parts match perfectly learned predicates. CIG is given by $CIG = 1 - MCE_{model}/MCE_{rand}$ normalized between $[0, 1]$. See Appendix I for more details.

## 4 Method

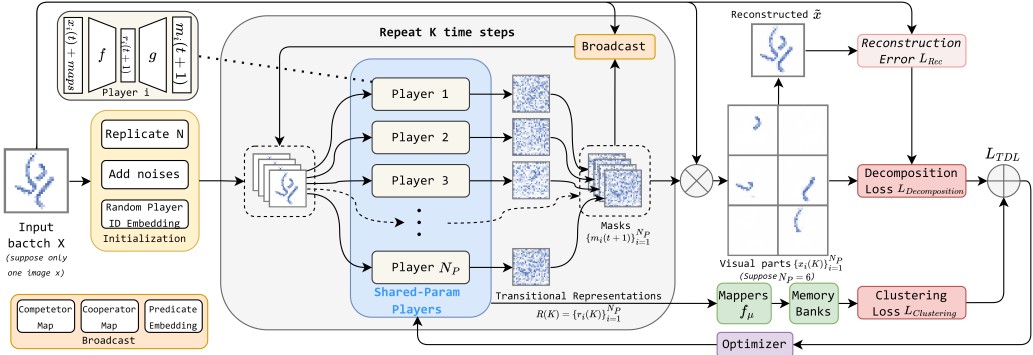

Figure 2: Overview of our architecture. The decomposition process takes $K$ steps to iteratively refine the generated visual parts. $N_P$ models generate in parallel, each generates one part and communicates through "Broadcast". The mapped representations will be stored in a memory bank for clustering.

To implement the TDL, we need an encoder $f : x \to R$ that decomposes the input $x$ into $R = \{r_i\}_{i=1}^{N_P}$ where each $r_i$ decoded by a decoder $g : r_i \to m_i$ into a visual part $x_i = m_i x, m_i \in [0, 1]^{H \times W \times C}$ is a mask. Inspired by Wu et al. (2022), we adopt a U-Net-based diffusion model Song et al. (2021) to iteratively refine the generated mask, the encoder downsamples the $x$ to the embedding $r_i$ which upsampled by the decoder as $m_i$. The model iterates $K$ steps. $N_P$ copies of the model sharing the same parameters, generate $N_P$ masks in parallel with each produces one mask. At each step, for model $i$, the mask $m_i(t)$ generated in the previous step (a random mask for the first step) and other feature maps, such as other models' output, are inputted, to produce an updated mask $m_i(t + 1)$. After $K$ steps, the model outputs $N_P$ visual parts and the corresponding representation $R$.

To generate multiple meaningful visual parts at the same time, we propose a game-theoretic method in Section 4.1 inspired by Gemp et al. (2021) who model PCA as a competitive game of principal components. We regard the decomposition process as a cooperative game of visual parts that converge in $K$ steps to cooperatively reconstruct the input while competing with each other to avoid repetition and so on. Each part is adjusted by a "player", one of the $N_P$ copies of the model. With the visual parts generated, we use prototype clustering to implement the EM algorithm in Section 4.2. We also introduce a shape score to measure model performance and serve as a reward to tune the unsupervised learned model with reinforcement learning in Section 4.3. Figure 2 shows an overview of our method.

## 4.1 GAME-THEORETIC DECOMPOSITION

A player model adjusts the generated visual parts to maximize the utility modeled by a GT loss

$$L_{GT} = L_{Rec} + \alpha_1 L_{overlap} + \alpha_2 L_{resources} + \alpha_3 L_{norm} \tag{4}$$

evaluates the equilibrium state composed of $N_P$ generated parts $\overrightarrow{x} = (x_1, x_2, ...x_{N_P})$, in detail:

**Reconstruction Error**. $L_{Rec}$ evaluates the reconstructed input $\tilde{x} = \sum_i^{N_P} x_i$ using a combination of focal loss (Lin et al., 2017) and dice loss (Milletari et al., 2016).

**Overlapping Penalty**. $L_{overlap} = \sum_{H,W,C} \max(0, \tilde{x} - x)$ introduces a competition between players that avoids overlap between parts by penalizing redundant parts.

**Resources Penalty**. $L_{resources} = \sum_i^{N_P} max(0, q_R - |m_i|)$ where $q_R \in \mathbb{R}$ is the quota of a player, prevents one player from reconstructing everything while others output empty. The quota restricts one player from having enough resources to output the entire input, thus requiring cooperation.

**L2 Norm**. $L_{norm} = \sum_i^{N_P} ||\bar{m}_i||_2^2$ where $\bar{m}_i$ is the unactivated mask before input into the $Sigmoid$ function, which can simplify the search space of the model to accelerate convergence.

See Appendix E for further details. We follow SMLD (Song et al., 2021) to train a scoring network $\nabla_{m_i(t)} L_{GT} = f_S(s_i(t); \theta, \phi)$, where $\phi = \{\phi^i\}_{i=1}^{N_A}$ are prototype dictionaries from 1 to $N_A$ arities that will be discussed in Section 4.2, to approximate the gradient of optimal move for each player $i$ that maximizes utility $-L_{GT}$. The input state $s_i(t) = (e_i(t), \bar{x}_i(t))$ includes the feature map $\bar{x}_i(t) = concat(x; m_i(t); \sum_{k \neq i} m_k(t), ...)$ that contains the input, the current mask, and the moves of other players (i.e., "broadcast"), and an embedding $e_i(t) = e_i^{time}(t) + e_i^{pid} + e_i^{pred}(t)$ covers a time step $e_i^{time}(t) \in \mathbb{R}^{d_{emb}}$ and player index $e_i^{pid} \in \mathbb{R}^{d_{emb}}$ from learned embedding tables, and a predicate embedding $e_i^{pred}(t) = \sum_{\sigma \in \phi^1} P(\sigma|x_i(t))\sigma$ computed for every step.

The move sampled by Langevin dynamics $m_i(t+1) = m_i(t) + \epsilon \nabla_{m_i(t)} L_{GT} + \sqrt{2\epsilon} z(t), z(t) \sim N(0, I), t = 0, 1, ..., K$ where $\epsilon$ is the step size. A loss term $L_{SMLD}$ from the SMLD paper can be used to minimize $E[||\nabla_{m_i(t)} L_{GT} - f_S(s_i(t); \theta)||^2]$ to train the scoring network to give good approximations. We apply this term as a regularization to form **Decomposition Loss** $L_{Decomposition} = L_{GT} + \beta L_{SMLD}$ in Figure 2. Further details can be found in the Appendix A.

## 4.2 ONLINE PROTOTYPE CLUSTERING

We cluster $R$ to implement the EM algorithm in Equation 3. As discussed in Section 3, we learn multiple dictionaries for different arities. Each dictionary is composed of prototypes $\phi^i \in \mathbb{R}^{N_{\phi^i} \times d_\phi}$ where $N_{\phi^i}$ is the dictionary size and $d_\phi$ is the dimension of the prototype. We train a predicate head for each dictionary; for example, for 1-ary, $\mu_i^1 = f_\mu^1(r_i)$ maps a neural logic variable $r_i$ to a representation $\mu_i^1 \in \mathbb{R}^{d_\mu}$, for 2-ary, $\mu_k^2 = f_\mu^2(r_i, r_j)$ maps a pair $(r_i, r_j)$. In our work, mappers are implemented as convolution layers on visual parts and their combinations, e.g., $\mu_k^2 = f_\mu^2(x_i + x_j)$.

We perform an online clustering during training by maintaining a FIFO memory bank $M = \{M^i\}_{i=1}^{N_A}$ where $M^i \in \mathbb{R}^{L_M^i \times d_\mu}$ that adds new terms $\overrightarrow{\mu^i}$ after each training step. Similar to Caron et al. (2018), we run K-Means for every one or a few training steps after the warm-up epochs, in a set $\overrightarrow{\mu^i} + M^i$ for each dictionary, where, taking 1-ary as an example, $\overrightarrow{\mu^1} = (\mu^{1(1)}, \mu^{1(2)}, ..., \mu^{1(B)})$ is the set of 1-ary representations in an input batch of length $B$. $\overrightarrow{\mu^i}$ has gradients while $M^i$ does not. We drop unwanted terms, such as empty ones, and randomly sample, for example, 30% pairs to increase efficiency.

The K-Means output assignments $C^i$ for each term in $\overrightarrow{\mu^i}$, we make pseudo-labels $Y^i$ for prototypes in the dictionary $\phi^i$ by assigning them to their nearest clustering centroids from $C^i$. Then we minimize the distance between prototypes and their assigned centroids in latent space by Cross-Entropy (CE) loss $\min_{\phi^i, \theta} L_{CE}^i = CELoss(dist(\overrightarrow{\mu^i}, \phi^i), Y^i)$ where $dist$ is a distance metric (e.g. L2 distance). The **Clustering Loss** is the sum of the CE loss for all dictionaries $L_{Clustering} = \gamma \sum_{i=1}^{N_A} L_{CE}^i$. We optimize it with the decomposition loss as $L_{TDL} = L_{Clustering} + L_{Decomposition}$ to implement the

TDL framework. More details can be found in the Appendix A.2. We visualize the latent space of a model trained in LineWorld for all the decomposed parts of the test set in Appendix D by tSNE, where we can see that the predicates in the dictionary are learned as different clusters, while the baseline, UPD, does not provide predicate information and shows a less organized latent space.

### 4.3 REINFORCEMENT LEARNING AND SHAPE SCORE

We employ PPO (Schulman et al., 2017) to tune an unsupervised learning model by regarding the decomposition process as an episode. We use a heuristic **shape score** to evaluate the decomposed part by 3 factors. (1) Continuity $R_{cont}$: the shape is not segmented and is an integral whole. $R_{cont} = \frac{max(A_C)}{sum(A_C)}$ where $A_C$ is the list of contoured areas for *segments* in a part. $R_{cont} = 1$ if the part is not segmented. We use $findContours$ in OpenCV to obtain the segments for 2D data and DBSCAN for 3D after converting to point clouds. (2) Solidity $R_{solid}$: no holes inside a part. $R_{solid} = \frac{A_P}{sum(A_C)}$ where $A_P$ is the space or area of the part. $R_{solid} = 1$ if there is no hole. (3) Smoothness $R_{smooth}$: the surfaces or contours of the part are smooth. $R_{smooth} = \frac{\rho_S}{\rho_O}$, where $\rho_S$ is the perimeter of the smoothed largest contour and $\rho_O$ is for the original contour. We apply RDP to smooth 2D data and alpha shape for 3D. The shape score $R_S = R_{cont} \times R_{solid} \times R_{smooth}$ normalized between 0 and 1. It can also be used to measure model performance. See Appendix J for more details.

## 5 EXPERIMENTS

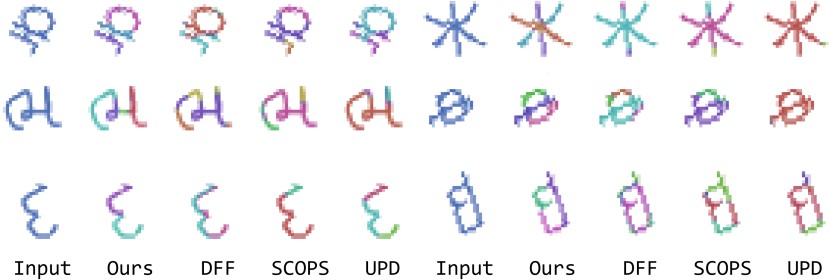

Figure 3: Examples from OmniGlot test set. Our method generates multiple interpretable strokes to reconstruct the input hand-written characters. As a comparison, the baseline methods segment the input into colored parts that are not valid strokes revealing a failure in learning compositionality.

In Section 5.1, we present our experiment setup to assess whether models can learn meaningful components in abstract objects without supervision, results discussed in Section 5.2. We then evaluate the learned representations by pre-training the models in an unsupervised manner and fine-tuning them for downstream tasks in Section 5.3. Finally, we conduct a human study in Section 5.4.

### 5.1 EXPERIMENT SETTING

We use three abstract compositional visual object datasets as shown in Figure 1, where parts cannot be separated by edges, features, colors, etc. Such contiguous shapes can only be decomposed by knowledge of compositionality, thus excluding confoundings. **LineWorld** is generated by the babyARC engine (Wu et al., 2022), consisting of images with 1 to 3 non-overlapping shapes made up of parallel or perpendicular lines. **OmniGlot** (Lake et al., 2015) contains handwritten characters. **ShapeNet5** is composed of 3D shapes in 5 categories (bed, chair, table, sofa, lamp) from ShapeNet (Chang et al., 2015) voxelized by binvox (Min, 2004 - 2019). We replace 2D conv layers with 3D when using this dataset. We create three downstream tasks based on them in Section 5.3.

We compare three state-of-the-art unsupervised part segmentation methods: **DFF** (Collins et al., 2018) clusters pixels by non-negative matrix factorization (NMF) on the activations of the last conv layer; **SCOPS** (Hung et al., 2019) and **UPD** (Choudhury et al., 2021) learn to produce a $k$ channels heatmap of parts self-supervisedly. The baselines require pre-trained visual backbones. Following

(Choudhury et al., 2021), we use VGG19 for 2D data and MedicalNet Chen et al. (2019), a ResNet-based high-resolution 3D medical voxel model, for 3D. We conducted hyperparameter searches for baselines to get their best results. Further details of the setting are provided in the Appendix B.

## 5.2 Unsupervised Learning of Transitional representation

|      | LineWorld | | | OmniGlot | | | ShapeNet5 | | | LW-G | | OG-G |
| --- | --- | --- | --- | --- | --- | --- | --- | --- | --- | --- | --- | --- |
|      | IoU | CIG | SP | MAE | CIG | SP | IoU | CIG | SP | IoU | Acc. | IoU |
| AE   | 97.7 | | - | 0.9 | | - | 85.1 | | - | | - | - |
| DFF  | - | 33.1 | 38.3 | - | 36.9 | 33.3 | - | 20.1 | 19.2 | 43.1 | 28.8 | 42.8 |
| SCO. | - | 35.7 | 42.4 | - | 38.6 | 38.9 | - | 23.1 | 24.3 | 46.8 | 26.4 | 46.9 |
| UPD  | - | 36.3 | 42.8 | - | 42.8 | 37.4 | - | 25.4 | 22.6 | 46.2 | 28.7 | 48.9 |
| **Ours** | **94.3** | **58.0** | **82.6** | **1.8** | **68.5** | **77.6** | **79.8** | **54.6** | **60.1** | **78.4** | **74.8** | **75.9** |
| **w/o RL** | 93.7 | 57.0 | 71.9 | 2.0 | 65.1 | 68.0 | 78.8 | 52.9 | 54.4 | 78.2 | 74.3 | 75.1 |

Table 1: Results on unsupervised learning and symbol grounding. SP represents the shape score. MAE and IoU for unsupervised learning are reference-only for comparison with a reference AutoEncoder (AE). Ours and w/o RL are our models with and without RL tuning. We compare DFF (Collins et al., 2018), SCOPS (SCO.) (Hung et al., 2019), UPD (Choudhury et al., 2021).

The results are presented in the first three columns of Table 1. We train Auto-Encoders as a reference to see if the generated parts match the input and whether the transitional representation preserves high-dimensional information. The LineWorld and ShapeNet5 inputs are binary, so we use IoU for a better intuitive. CIG is introduced in Section 3.3 and the shape score (SP) is discussed in Section 4.3.

Our model significantly outperforms the baselines with 58.0, 68.5, 54.6 CIG, and 82.6, 70.6, 60.1 SP in the three datasets, respectively. Even without reinforcement learning, the advantages remain. The low reconstruction error of 94.3 IoU, 1.8 MAE, and 79.8 IoU indicates the preservation of high-dimensional information. This is because the baselines depend on concrete visual features such as edges, colors, textures, etc., enabled by pre-trained vision backbones, to identify the boundaries of parts, which are absent in our datasets. For instance, there is no explicit color or texture difference between strokes in a handwritten character, and seems like contiguous integrity, thus can only be distinguished by the knowledge of strokes, which is learned via discovering compositional patterns. Figure 3 shows a comparison in the OmniGlot test set. See more samples in the Appendix M.

## 5.3 Adapt to downstream tasks

|        | Bed | | | Lamp | | | Sofa | | | Table | | |
| --- | --- | --- | --- | --- | --- | --- | --- | --- | --- | --- | --- | --- |
|        | IoU | CIG | SP | IoU | CIG | SP | IoU | CIG | SP | IoU | CIG | SP |
| w/ PT  | 67.3 | 48.1 | 52.9 | 61.1 | 42.1 | 49.1 | 62.2 | 46.8 | 45.2 | 68.3 | 50.1 | 54.6 |
| w/o PT | 18.1 | 19.0 | 13.2 | 18.3 | 19.9 | 14.6 | 21.5 | 18.9 | 19.8 | 19.9 | 22.1 | 17.9 |

Table 2: Transfer learning for our method on the ShapeGlot setup. "PT" means Pre-Training.

**Symbol Grounding.** We design two symbol grounding tasks: **LW-G** and **OG-G**. LW-G synthesized with babyARC while preserving the shape masks (e.g. lines) and the pair-wise relation annotations (e.g. perpendicular and parallel) from the engine as labels. The goal is to predict the shape masks and classify the pair-wise relations. We aligned the predicted mask with the ground truth by the assignment with minimal overall IoU before computing the metrics. We pre-train the models on LineWorld and add a relation prediction head on the top of baselines while our method directly adapts the 2-ary predicate head for relations classifying. OG-G is a subset of OmniGlot with the provided stroke masks as ground truth to predict. We align prediction and ground truth as LW-G. We pre-train models on OmniGlot without OG-G samples. We show examples of LW-G and OG-G in Appendices

G and 9. As shown in Table 1 under LW-G and OG-G, we achieve 78.4 IoU, 74.8 Acc. in LW-G and 75.9 IoU in OG-G, outperforming baselines whose relation prediction did not converge due to incorrect segmentations. This demonstrates that the learned transitional representation enables smooth transfer to a concrete set of symbols, as hypothesized in Section 3.1.

**Transfer Learning.** Following ShapeGlot (Achlioptas et al., 2019), we pre-train with shapes from "chair" and other 4 similar categories, then transfer to 230∼550 samples from unseen categories "Bed", "Lamp", "Sofa", "Table". We compare our method with and without pre-training. The results in Table 2 demonstrate that the learned representations are reusable and effectively generalized to unseen classes. Without pre-training, the samples for each class were not sufficient to converge.

## 5.4 HUMAN EVALUATION

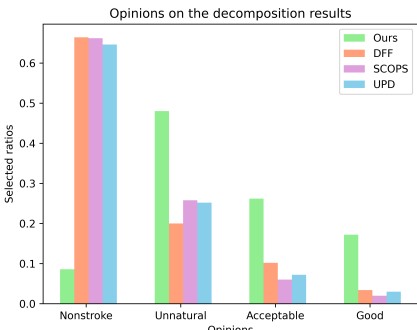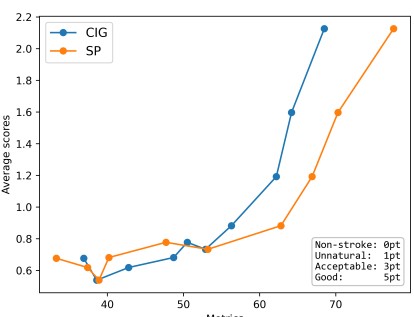

Figure 4: Left: Results of the human evaluation. Right: The qualitative scores compared to metrics.

**Human Interpretability.** We conduct a human evaluation to evaluate our method and the baselines by humans. We randomly selected 500 samples from the OmniGlot test set and the decomposition results for each method. We use the Google Vertex AI (Google, 2021) data labeling service to evaluate the results as an image annotation task with three annotators. Annotators are given decomposed samples and asked to provide one of four opinions, examples of which can be found in Appendix K, as outlined in an instruction that must be read before the task begins. The 2000 samples are shuffled and then randomly assigned to the annotators. The results in Figure 4 left show a much better interpretability of our method, while ∼ 65% of the baseline results are not considered strokes.

**Interpretability vs. Metrics.** We further train 6 more models, in addition to the 4 models in Section 5.2, to get near-even distributed SP and CIG by early stop. We then conduct human evaluations in the same way as above. We assign points for each sample by `Non-stroke:0`, `Unnatural:1`, `Acceptable:3`, `Good:5`, then average them as scores for each model. And compare the scores with the metrics of each model in Figure 4 right, which shows that SP and CIG are positively correlated with human interpretability, which can be used as reliable predictors of interpretability.

## 6 CONCLUSION

This paper presents the TDL framework, which uses an EM algorithm to learn a neural-symbolic transitional representation that incorporates structural information into representations. We introduce a game-theoretic diffusion model with online prototype clustering to implement TDL and assess by proposed metrics, clustering information gain, and shape score. We evaluate our method on three abstract compositional visual object datasets, using unsupervised learning, downstream task experiments, and human assessments. Our results demonstrate that our method largely outperforms existing unsupervised part segmentation methods, which rely on visual features instead of discovering compositionality. Furthermore, our proposed metrics are in agreement with human judgment. We believe that our work can help bridge the gap between neural and symbolic intelligence.

# 7 REPRODUCIBILITY STATEMENT

To guarantee the reproducibility and completeness of this paper, we provide the full details of our model architecture and implementation in the Appendix A. Appendix B contains information about the generation or preprocessing of samples for each dataset and the split used in each experiment. Appendix C contains our settings for hyperparameter search and our hardware platform information. The tricks we used to calculate the GT loss are included in Appendix E. We also make our code and data publicly available for readers to reproduce our work.

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

# A ARCHITECTURE DETAILS

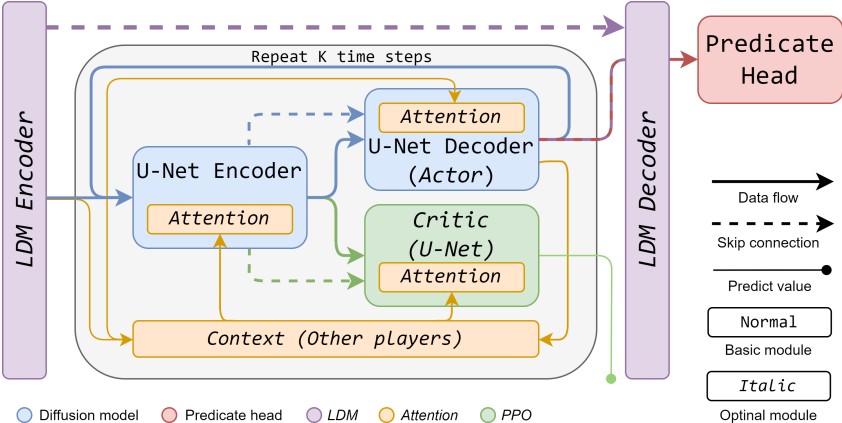

Figure 5: Illustration of the model architecture. Blocks with italic font are optional optimizations. Different color marks the information flow of the 5 different modules.

We present the details of our model here. The architecture is depicted in Figure 5, which includes two core components, a diffusion model, and a predicate head, as well as three optional optimizations: Latent diffusion model (LDM) to improve efficiency, an attention mechanism to incorporate context information such as the moves of other players, and PPO to tune the model. We assume a single 2D input by default in this section for simplicity, and it is easy to extend to batch cases.

## A.1 DIFFUSION MODEL

---

**Algorithm 1** Decompose an input

---

**Require:** an scoring model $f_S(\cdot; \theta, \phi)$, decoder $f_g : \mathbb{R}^d \mapsto \mathbb{R}^{H \times W \times C}$ map representation to visual part

**Require:** an input $x \in \mathbb{R}^{H \times W \times C}$, number of players $N_P$, time steps $K$

  1: Randomly initialize variables of $N_P$ players $R(0) \in \mathbb{R}^{N \times d}$
  2: Initialize player id embedding $e^{pid} \in \mathbb{R}^{N \times d_{emb}}$ and time embedding $e^{time} \in \mathbb{R}^{K \times d_{emb}}$
  3: **for** $t \in \{1, ..., K\}$ **do**
  4:     $\overrightarrow{x}(t) = f_g(R(t))$, where $\overrightarrow{x}(t) \in \mathbb{R}^{N \times H \times W \times C}$       ▷ visual parts of $N$ players
  5:     $\forall j \in \{1, ..., N_P\}, \psi_j(t) = [x; x_j(t); \sum_{k \neq j} x_k(t); \sum_{k \neq j, \sigma \in \phi^2} P(x_k(t), \sigma | x_j(t)) x_k(t)]$
  6:     $\forall j \in \{1, ..., N_P\}, e_j(t) = e_j^{pid} + e_j^{time}(t) + GetPrediEmb(\psi_j(t), \phi)$     ▷ see A.2.1 for $GetPrediEmb$
  7:     $R(t+1) = f_S([\psi(t), e(t)]; \theta, \phi)$
  8: **end for**
  9: **return** $R(K)$

---

The algorithm 1 illustrates how the model decomposes an input. Notice that there are a few differences between Section 4.1 where the scoring model also incorporates the prototype dictionaries. Our model is based on SMLD (Song et al., 2021). SMLD adds a dense layer between each convolution layer in U-Net, which introduces the time-step embedding $e^{time} \in \mathbb{R}^{d_{emb}}$ by adding the output of this dense layer to the output of the convolutional layer. At each time step $t$, for a player $j$, the input state $s_j(t)$ composed of feature maps $\psi_j(t) \in \mathbb{R}^{H \times W \times C_{inp}}$ and the embedding $e(t) \in \mathbb{R}^{d_{emb}}$, to produce updated decomposed composition $x_j(t+1)$.

## A.2 PREDICATE HEAD

The predicate head embeds decomposed parts into a latent space and clusters them to learn dictionaries for predicates. The head $f_\mu^i$ for $i$-ary predicates implemented with a shared mapper $f_{mapper}$ for

all arities implemented as convolution layers with global pooling, mapping part $x_j \in \mathbb{R}^{H \times W \times C}$ to tensor $\tau_j \in \mathbb{R}^{d_{mapper}}$, and an arity-specific linear layer $Q^i \in \mathbb{R}^{d_{mapper} \times d_\mu}$ map the $\tau_j$ to embedding $\mu_j^i$. In our work, we apply the multi-prototype trick (Yang et al., 2018), which means that we apply $K_{\phi^i}$ prototypes for each $i$-ary cluster, resulting in a multi-prototype dictionary $\phi^i \in \mathbb{R}^{K_{\phi^i} \times N_{\phi^i} \times d_\mu}$.

---

**Algorithm 2** Compute clustering error for 1 and 2-ary predicates

---

**Require:** $L$ variables $R \in \mathbb{R}^{L \times d}$, prototypes $\phi^1 \in \mathbb{R}^{N_{\phi^1} \times d_\mu}$, $\phi^2 \in \mathbb{R}^{N_{\phi^2} \times d_\mu}$
**Require:** memory banks $M^1 \in \mathbb{R}^{L_M^1 \times d_\mu}$, $M^2 \in \mathbb{R}^{L_M^2 \times d_\mu}$
**Require:** mappers $f_\mu^1, f_\mu^2 : \mathbb{R}^d \mapsto \mathbb{R}^{d_\mu}$, decoder $f_g : \mathbb{R}^d \mapsto \mathbb{R}^{H \times W \times C}$
**Require:** distance metric for two vector sequences of length $n_1$ and $n_2$: $d : \mathbb{R}^{n_1 \times d_\mu} \times \mathbb{R}^{n_2 \times d_\mu} \mapsto \mathbb{R}^{n_1 \times n_2}$

1: $\overrightarrow{x} = filter(f_g(R))$, where $\overrightarrow{x} \in \mathbb{R}^{\mu' \times H \times W \times C}$  $\quad\triangleright$ filter out unwanted visual parts
2: $\overrightarrow{\mu^1} = sample(f_\mu^1(\overrightarrow{x}), N_\mu^1)$, where $\overrightarrow{\mu^1} \in \mathbb{R}^{N_\mu^1 \times d_\mu}$  $\quad\triangleright$ random sample $N_\mu^1$ items
3: $\overrightarrow{p} = filter(\{x^q + x^w, \forall x^q, x^w \in \overrightarrow{x}\})$, where $\overrightarrow{p} \in \mathbb{R}^{\mu'' \times H \times W \times C}$  $\quad\triangleright$ pair representations
4: $\overrightarrow{\mu_2} = sample(f_\mu^2(\overrightarrow{p}), N_\mu^2)$, where $\overrightarrow{\mu^2} \in \mathbb{R}^{N_\mu^2 \times d_\mu}$  $\quad\triangleright$ random sample $N_\mu^2$ items
5:
6: **function** ASSIGN($C^{i'}, \phi^i$)
7: $\qquad \bar{C} = \{\frac{\sum_{0 < j \le N_\mu^i} \phi_j^i \mathbb{K}(C_j^i == k)}{\sum_{0 < j \le N_\mu^i} \mathbb{K}(C_j^i == k)}, \forall 0 < k \le N_{\phi^i}\}$  $\quad\triangleright$ Centroids for $i$-ary cluster
8: $\qquad d_E = L1\_Dist(\bar{C}, \mathbb{I}_{N_{\phi^i}}), d_E \in \mathbb{R}^{N_{\phi^i} \times N_{\phi^i}}$ $\triangleright$ L1 Distance from centroids to permutations of assignments
9: $\qquad$ **while** $k < N_{\phi^i}$ **do**
10: $\qquad\qquad (row, col) = argmin(d_E)$
11: $\qquad\qquad Y_k^i = col$
12: $\qquad\qquad d_E[row, :] = +inf$
13: $\qquad\qquad d_E[:, col] = +inf$
14: $\qquad$ **end while**
15: $\qquad$ **return** $Y^i$
16: **end function**
17:
18: **function** GETLOSS($\overrightarrow{\mu^i}, M^i, \phi^i$)
19: $\qquad C^i = K\text{-}Means([\overrightarrow{\mu^i}; M^i]), C^{i'} = \{C_j^i, j = \{1, ..., N_\mu^i\}\}$
20: $\qquad Y^i = Assign(C^{i'}, \phi^i), Y^i \in \mathbf{Z}^{N_\mu^i \times N_{\phi^i}}$
21: $\qquad dist^i = d(\overrightarrow{\mu^i}, \phi^i), dist^i \in \mathbb{R}^{N_\mu^i \times N_{\phi^i}}$
22: $\qquad$ **return** $CrossEntropyLoss(dist^i, Y^i)$
23: **end function**
24: **return** $GetLoss(\overrightarrow{\mu^1}, M^1, \phi^1) + GetLoss(\overrightarrow{\mu^2}, M^2, \phi^2)$

---

In each time step $t$, for player $j$, the predicate head accepts the current decomposition $x_j(t)$ as input and outputs the predicate embedding and cooperator map. After $K$ time steps, given the representation $R$, the predicate head computes the clustering error by Algorithm 2. We further explore a **Higher-Order Logic (HOL)** predicate, an HOL representation is given by dot-product attention, as $h_j^i = \sum_{k \ne j} P(r_j^i, r_k^i) r_k^i$, and HOL pairs $\eta_{q,w}^i$ can be constructed with $h^i$ in the same way as 2-ary predicates by summing up the corresponding image parts.

### A.2.1 PREDICATE EMBEDDING

For a player $j$, the predicate embedding is computed as $e_j^{pred}(t) = \sum_{\sigma \in \phi^1} P(\sigma|x_j(t))\sigma$ where $P(\sigma|x_j(t)) \propto -dist(\sigma, q_j^1(t))$, $dist$ is a distance metric, we use L2 distance by default in this work and $q_j^1(t) = Q^1 f_{mapper}(x_j(t))$. It represents the potential 1-ary predicate of the current decomposition result. If using a higher-order predicate, we sum $e_j^{pred}(t)$ with $\sum_{\sigma \in \phi^{h1}} P(\sigma|h_j(t))\sigma$, where $P(\sigma|h_j(t)) \propto -dist(\sigma, q_j^{h1}(t))$, $q_j^{h1} = Q^{h1} f_{mapper}(f_g(h_j(t)))$ and $h_j(t) = \sum_{k \ne j} P(r_j, r_k) r_k$ where $P(r_j, r_k) \propto r_j r_k$, the dot-product distance, to model relations over grouped players.

## A.3 LATENT DIFFUSION MODEL

LDM (Rombach et al., 2022) uses an Auto-Encoder to encode an input $x \in \mathbb{R}^{H \times W \times C}$ into a compressed input $x' \in \mathbb{R}^{H' \times W' \times C'}$, where $H' < H$ and $W' < W$, and input the compressed input $x'$ instead of the original input $x$ into the diffusion model, which outputs the composition $x_{j'} \in \mathbb{R}^{H' \times W' \times C'}$ in latent space, then uses the decoder to decompress it into the original pixel space $x_j \in \mathbb{R}^{H \times W \times C}$. We use a U-Net-based Auto-Encoder in our work, and we keep the skip connection that inputs the downsampled middle results from the encoder to the decoder.

## A.4 ATTENTION LAYERS

We introduce optional transformer blocks (Vaswani et al., 2017) between the convolution layers in U-Net, similar to CLIP (Radford et al., 2021). There are two types of transformer blocks that have been used in our methods, self-attention, and cross-attention, self-attention is only used in the encoder part when using an LDM since there is no context, and all other places use the cross-attention. For cross-attention, the context is the set of the mapped representation of the competitors $\{\tau_1, \tau_2, ..., \tau_{j-1}, \tau_{j+1}, ...\tau_m\}$ for player $j$ mapped by $f_{mapper}$ introduced in Section A.2. We utilize memory-efficient attention in xFormers (Lefaudeux et al., 2022) in our implementation.

Attention also gives a powerful tool for incorporating multimodal information (Cheng et al., 2021c) and inductive biases (Cheng et al., 2021a;b), the context may come from other modalities, similar to Rombach et al. (2022), and in a multi-agent case, the context could come from other agents.

## A.5 PPO AND ACTOR-CRITIC

PPO (Schulman et al., 2017) uses an Actor-Critic framework to train the agent, we apply a shared encoder actor and critic that the U-Net encoder in the diffusion model is shared, and we train two decoders for the actor and critic, respectively. The actor samples a move with a Bernoulli distribution, where the probability is given by the diffusion model output, on each pixel as a mask. A reward function rating on this mask by the loss and the shape score. And the critic is trained to predict the reward given a state. We follow the PPO algorithm with the implementation of PPO2 (OpenAI, 2018-2021). The model is updated every few steps of sampling. A small buffer that saves actions, states, following states, and other useful information is maintained and retrieved iteratively when the model is updated.

## B DATASET DETAILS

We present statistical information about the data we used, the division of the training, testing, and development sets, and the details of how we generated and pre-processed the datasets.

**LineWorld.** We employ the babyARC engine (Wu et al., 2022) to generate the LineWorld dataset. This dataset consists of objects made up of lines, which are related to each other in terms of parallelism and perpendicularity. Each sample is an image containing between one and three "Lshape", "Tshape", "Eshape", "Rectangle", "Hshape", "Cshape", "Ashape", and "Fshape" objects, each of which may have a different size and one of three randomly selected colors. The shapes are non-overlapping and placed randomly on a white background. In total, we synthesize 50000 samples, which are divided into 8:1:1 splits for training, development, and testing.

**LineWorld-Grounding (LW-G).** We used the babyARC engine to synthesize the LW-G dataset. The objects in LW-G are composed of lines as a basic concept with four relations to each other:

**Parallel.** The two lines are parallel.

**VerticalMid.** The two lines are vertical, with an endpoint of one line attached to the middle of another.

**VerticalEdge.** The two lines are vertical, with an endpoint of one line attached to an endpoint of another.

**VerticalSepa.** The two lines are vertical, but the endpoint of one line is not attached to another.

Each object is composed of a pattern. Apart from the 4 relationships as the basic pattern that simply samples two random lines following the 4 relations, we design 7 more complex patterns:

| Patterns | Sampling process |
|---|---|
| F pattern | Sample line in a random position, sample the second line that "VerticalMid" to the first line, then sample a third line "VerticalEdge" to the first |
| E pattern | Sample one more line that "VerticalEdge" to the first line in a "Fpattern" with one endpoint attached to the unattached endpoint of the first line |
| A pattern | Sample one more line that is "Parallel" to the first line in a "Fpattern", the line can be anywhere that does not overlap with the first line |
| C pattern | Sample one line in a random position, sample the second line that "VerticalEdge" to the first line, then sample a third line "VerticalEdge" to the first but attach to another endpoint |
| H pattern | Sample one line in a random position, sample the second line that "VerticalMid" to the first line, then sample a third line "Parallel" to the first |
| P pattern | Sample one more line that "VerticalEdge" to the second line in a "Fpattern" on the unattached endpoint |
| Rect | Sample one more line that "VerticalEdge" to the second line in a "Cpattern" on the unattached endpoint |

The length and direction of the lines are randomly determined and the color of each line is randomly selected from two available colors. This implies that an "F pattern" does not necessarily have to be in the form of an "F" - the two parallel lines may have different directions and lengths, and this is true for all other patterns as well. Each sample is composed of an image of one object, a list of concepts (i.e., lines) where each concept is represented by a mask that points out this concept in the image, and a list of relation tuples between the concepts (e.g. $(line1, line2, parallel)$). In total, we generated 7000 samples, with a 5:1:1 split for train, dev, and test sets.

We assess concept prediction in this manner: Let us assume that the model provides $k$ potential concepts and $k^2$ connections between them, forming a complete graph $G_C$. We then determine whether $G$ is included in $G_C$. We assign $c$ to the closest candidates with the least Intersection over Union (IoU) and calculate the mean IoU. Subsequently, we calculate the top-1 accuracy of the predicted relation based on the assignment of nodes.

Our model reads the relation between two parts directly from the two-ary predicates. For baselines, we added a relation prediction head that takes two parts of the heatmap as input and predicts the relation between them. This prediction head has a similar structure to our composition mapper, which is used to embed the outputted composition from each player for clustering. The parts are mapped first, and then a classifier is used to predict the relation.

**OmniGlot.** The OmniGlot dataset was collected by Lake et al. (2015) using Amazon's Mechanical Turk (Amazon, 2005). They recruited participants to draw 1623 characters from 50 different alphabets, each of which was drawn 20 times by different people. Each sample was composed of multiple stroke sequences of $[x, y, t]$ coordinates that documented the strokes used to create the character. We employ the program from Lake et al. (2015) to transform the stroke sequences into images. We allocate 24000, 1500, and 1500 images for the training, testing, and development sets, respectively, and the remaining images are used for the OG-G dataset.

**OmniGlot-Grounding (OG-G).** OG-G is composed of the remaining 5811 samples, apart from the image of the character, each sample also contains a set of images of the strokes that are converted from the sequence of each stroke that composed the character as the ground truth. We use 4311, 750, and 750 samples for train, test, and dev sets.

**ShapeNet5.** ShapeNet5 consisted of all 20938 shapes of 5 categories (bed, chair, table, sofa, lamp), suggested by Achlioptas et al. (2019) that they are composed of shared basic elements, from ShapeNetCore v2 (Chang et al., 2015), an updated version of the core ShapeNet dataset. There are

many ways to represent 3D data including point clouds, meshes, voxels, Signed Distance Fields (SDF), and octrees. In order to make 3D shapes directly applied to the same architecture with other datasets, we choose to voxelize the shapes; thus, we can handle them by simply replacing 2D convolutions with 3D. We use the binvox library (Min, 2004 - 2019; Nooruddin & Turk, 2003) to voxelize shapes into solid voxels (i.e., the interiors of the shapes are filled). We applied 15938, 2500, and 2500 samples for train, test, and dev sets.

**ShapeGlot Transfer Learning.** This transfer learning dataset consists of a pre-training set and fine-tuning sets. The pretraining set covers 11470 samples from 5 categories (chair, bench, cabinet, bookshelf, bathtub) that we regard as having similar basic compositions. Four fine-tuning sets correspond to four categories that share similar basic elements with the pretraining set: bed (233), lamp (532), sofa (550), and table (580), number of samples for each category is provided in brackets. We voxelize the samples the same way with ShapeNet5.

## C  HYPER-PARAMETER SEARCH

| Params | Distribution | Params | Distribution |
|---|---|---|---|
| $lr$ | $\{1e-3, 2e-3, 5e-4\}$ | $th_S$ | $U(0.1, 0.5; 0.05)$ or None |
| $K$ | $U(3, 9; 1)$ | quota | $U(8, 32; 4)$ |
| $\alpha_{overlap}$ | $U(0.1, 0.25)$ | $d_{emb}$ | $\{64,128,256,512\}$ |
| $\alpha_{resources}$ | $U(0.05, 0.2)$ | $d_{sampler}$ | $\{32,64\}$ |
| $\gamma_{cluster}$ | $U(5e-3, 2e-2)$ | $d_{mapper}$ | $\{128,256,512\}$ |
| $\sigma$ | $\{2.5,5,10,15,25\}$ | | |

Table 3: Hyper-parameter search distributions.

We use Weights & Biases Sweep (Biewald, 2020) to perform hyperparameter searches for our method. Table 3 shows the distribution of the empirically significant parameters in our hyper-param search. $\alpha_{overlap}$ and $\alpha_{resources}$ are $\alpha_1$ and $\alpha_2$ in Equation 4. $U(a, b; s)$ is a discrete uniform distribution with a step size of $s$ between $a$ and $b$, while $U(a, b)$ is a uniform distribution between $a$ and $b$. We use a combination of random search and grid search to explore the search space, with random search used to identify good traces, and then, with the help of the visualization tool provided by Sweep, we get a smaller search space for a finer grid search. The final good range determined by the random search can vary for different experiments; however, the distribution given by Table 3 provides the common initial range that empirically likely covers the optimal sets to explore.

We conducted our experiments on our internal clusters, and a major workload has the following configuration: six Quadro RTX 5000 GPUs and one Quadro RTX 8000 GPU, along with an Intel (R) Xeon (R) Silver 4214R CPU @ 2.40GHz and 386 GB RAM. We employed PyTorch Lightning (Falcon & The PyTorch Lightning team, 2019) for parallel training.

## D  LATENT SPACE VISUALIZATION

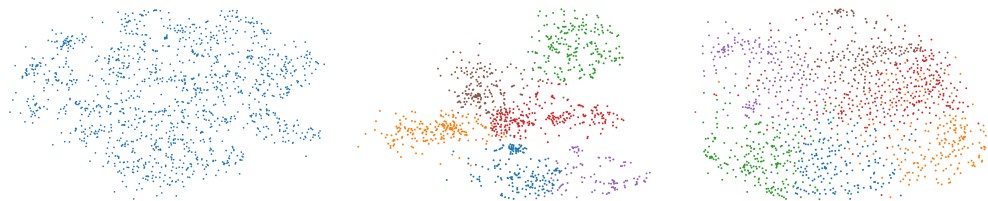

Figure 6: t-SNE for parts segmented by UPD (Choudhury et al., 2021) (left), and the latent space of 1 (mid) and 2-ary (right) embeddings of parts and pairs, colored by nearest predicates, decomposed by our method in the LineWorld test set. UPD provides no such predicate information thus not colored.

## E GT LOSS DETAILS

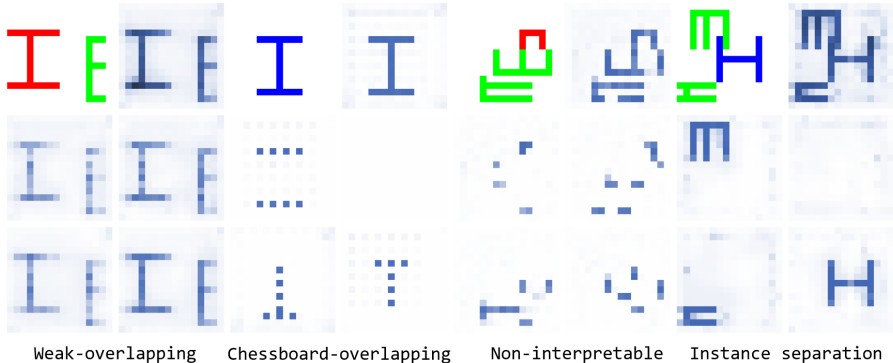

Weak-overlapping   Chessboard-overlapping   Non-interpretable   Instance separation

Figure 7: Unwanted cases were shown during our mechanism design. The first row shows pairs of input (left) and reconstructed result (right), and the remaining rows are the outputs of players. Another two common cases not shown here are all players output the same output and one player output while all others are empty.

There are some important tricks to apply apart from only using the GT loss to avoid some unwanted outputs as shown in Figure 7. The **weak-overlapping** is a tricky way learned by the model to bypass the mechanisms. It means that each player produces a weak copy of the input $x_j = cx$ where $0 < c < 1$, it fully meets the requirements of the GT loss. We solve it with a **soft step function** trick, which computing the overlapping loss by $max(0, \sum_j^{N_P} Step(x_j; th_S) - Step(x; th_S))$, where $Step$ is a point-wise step function that assigns 1 to a position if the value is above a threshold $th_S$ and 0 otherwise. A soft step function can be implemented with a Heaviside function. With a step function, the overlapping loss becomes more sensitive, and a small value in a position will be regarded as an occupation. And the threshold controls the sensitivity. Thus, the model cannot cheat with weak overlapping and be more careful when assigning values.

Another undesirable situation is **chessboard overlapping**, where players avoid overlap by outputting pixels with intervals. This is typically caused by using a deconvolutional upsampling (Odena et al., 2016) and can be solved by replacing it with bilinear interpolation. The **non-interpretable** shape occurs in a base model without a predicate head. By applying predicate clustering and other techniques that encourage interpretability, such cases can be largely reduced. In the **instance separation** case, the model learns an unwanted good case that gives a segmentation of instances. This case is caused by the given dictionary size being too large, thus the model can memorize all shapes.

## F ABLATION STUDY ON MODEL ARCHITECTURE

|  |  |  | LineWorld | | | OmniGlot | | |
|---|---|---|---|---|---|---|---|---|
|  |  |  | IoU | CIG | SP | MAE | CIG | SP |
| Base |  |  | 91.3 | 26.3 | 28.4 | 2.5 | 31.8 | 23.8 |
| ~ | + Cluster |  | 92.3 | 54.6 | 68.9 | 2.4 | 61.7 | 64.8 |
| ~ | ~ | + HOL | 92.7 | 56.9 | 71.5 | 2.0 | 64.7 | 67.5 |
| ~ | ~ | + PPO | 91.9 | 57.1 | 82.6 | 2.0 | 67.2 | 77.4 |
| ~ | ~ | + Attn. | 94.9 | 56.2 | 70.8 | 1.8 | 63.4 | 66.8 |
| ~ | ~ | + LDM | 91.7 | 53.6 | 66.9 | 3.2 | 59.9 | 62.8 |
| Full model |  |  | 94.3 | 58.0 | 82.6 | 1.8 | 68.5 | 77.6 |

Table 4: Ablation study results. "~" means repeating the above row.

We perform an ablation study of the proposed predicate clustering method, PPO tuning, and optimizations in LineWorld and OmniGlot datasets. The results are listed in Table 4.

"**Base**" means the model trained with only decomposition loss. Without clustering, the model gives an arbitrary decomposition that reconstructs the input while meeting the game mechanisms, which generate parts with diverse near-random shapes, thus having low CIG and SP.

"**+ Cluster**" means adding a clustering loss to the base model. With cluster loss, the model does dictionary learning, which significantly improves the CIG and SP, since the model learns to find common elements to represent the data. It also shows that learning an efficient dictionary itself also results in simpler and more natural shapes. Although not sufficient to learn the human interpretable shapes.

"**+ HOL**" adds the higher-order predicate optimization to the model with clustering, and it shows marginal improvement, which may be due to the low complexity of the shapes in our datasets, which do not contain too complex relationships that need to be depicted with higher-order predicates. And a better way of representing higher-order predicates may also lead to better results.

"**+ PPO**" adds a PPO tuning with heuristic reward to the model with clustering, and it clearly improves the SP in both datasets due to the introduction of an explicit bias that guides the model to learn more natural shapes, which shows the importance of feedback and environmental interactions.

"**+ Attn.**" adds attention layers to the model with clustering; it also gives marginal improvement which may be due to the relatively limited complexity of our dataset. Moreover, as discussed earlier, a better case for cross-attention is multimodal learning. And we do not test self-attention, which we regard should be applied to larger datasets.

"**+ LDM**" uses a latent diffusion in the model with clustering, the result shows that the use of LDM has only limited harm to the model performance; compared to efficiency improvement, the downside is quite acceptable.

In conclusion, the findings demonstrate that dictionary learning is a critical factor in learning transitional representation, while PPO can effectively enhance the representation. HOL and attention offer minor improvements in our datasets; however, they may be more beneficial in a more complex dataset and with better higher-order representation. LDM can improve model efficiency with minimal impact on performance, which is essential for scaling to larger inputs.

## G  HIERARCHICAL CONCEPTS AND RELATIONS

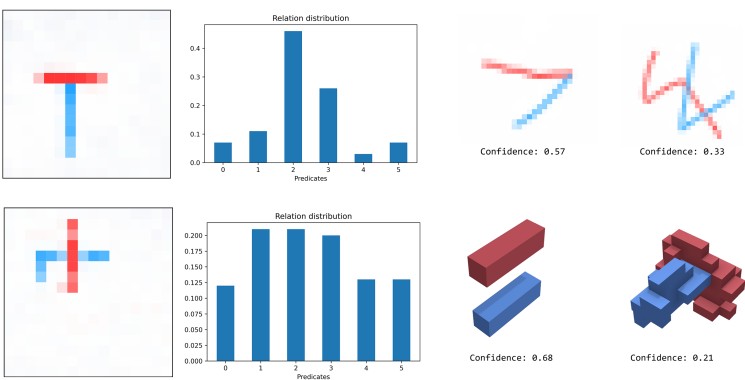

Figure 8: Illustration of a relation clustering in the three datasets.

We give more insights on how relation clustering or 2-ary predicates work here. In Figure 8, we show examples from the three datasets, we query a pair of red and blue parts in the learned 2-ary predicate dictionary by comparing the distance of their representation obtained by the mapper and query layer and the 2-ary predicate prototypes, which gives a confidence distribution over each predicate, or a relative distance from them. One can also use an absolute distance with a threshold to obtain a better

Out-of-Distribution (OOD) detection ability for not only relation prediction but also other dictionaries and also better robustness (Yang et al., 2018), for simplicity, we keep a relative distance in our work.

On the left side of the figure, we visualized the distribution in two LineWorld samples, where a more common pattern "T" is close to predicate 3, while another overlapping pattern that is not allowed in the dataset is remote to all prototypes. On the right side, we provide the confidence of the predicate with the largest confidence for each sample; we can see that the more common sample shows higher confidence than a more random one which is hard to be concluded as any categories.

While the 1-ary predicates learn visual primitives, the 2-ary predicates implicitly learn common combinations, and the higher-ary and order predicates can be seen as learning subparts. It shows that our method can implicitly learn the hierarchy concepts (Lake et al., 2015).

## H  SYMBOL GROUNDING

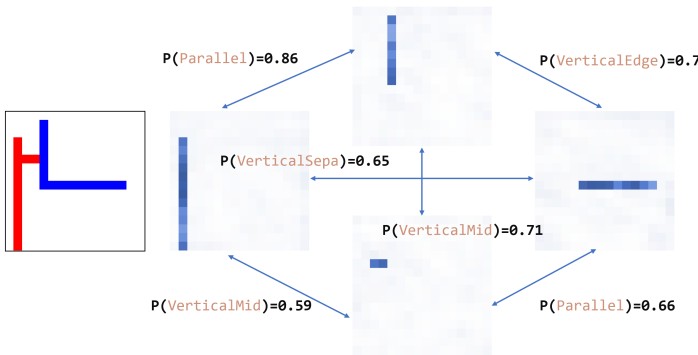

Figure 9: Illustration of the symbol ground of an LW-G sample. Empty players are omitted.

Here, we show how symbol grounding works that grounds an image to a set of predefined predicates. Figure 9 shows an example of grounding an LW-G sample to the predicates defined by the babyARC engine. The model first decomposes the image into parts, which are the lines in different positions with different lengths, as 1-ary predicates. Then predict the relationships between the pairs of the parts by the relation predictor which is implemented as 2-ary predicate prototypes.

This gives a complete graph $G_S$ with 1-ary concepts as nodes and 2-ary relationships as edges. Suppose the ground-truth graph is $G_{gt}$. Since the number of players is assumed to be larger than the number of ground-truth concepts, the model actually gives a complete graph with outputs from all players as nodes $G_P$, and we require $G_{gt}$ to be a sub-graph of $G_P$ and the nodes that are included in $G_P$ but not $G_{gt}$ to be empty. In training time, we extract the best matching subgraph of $G_P$ to $G_{gt}$ as $G_S$ and compute the loss. In inference time, the prediction is the complete graph of the non-empty nodes. To extend our method to a non-complete graph, we may simply introduce a threshold to relation prediction, or other OOD methods including using the absolute distance in prototype classifier as discussed in G, or simply introduce a category for empty relation.

## I  ANALYSIS OF CLUSTERING INFORMATION GAIN

To calculate the CIG of a model, we first decompose each sample $x$ in the test set into parts $\overrightarrow{x}$, resulting in the set $\overrightarrow{X}$ of all parts. We then generate a set of parts $\overrightarrow{X}_{rand}$ randomly sampling parts from samples in the test set, using a mask with a normal distribution in each position. We then compute the MCE of both sets to obtain $MCE_{model}$ and $MCE_{rand}$, which allows us to calculate the CIG. To do this, we reduce the dimension of the parts and then run a K-Means clustering.

The influence of dimension reduction on fairness can be seen by comparing three typical dimension-reduction techniques: PCA, Auto-Encoder (AE), and a pre-trained CNN, VGG19. Table 5 provides a comparison of the three methods. The "Reference" in LineWorld is composed of common elements

such as lines and two vertical lines in the mid or edge (i.e. "L" and "T" shapes) with different lengths, colors, positions, and directions; in OmniGlot, the "Reference" is the set of the ground-truth strokes. To test the stability of AE-based CIG, we trained AE ten times in each dataset and calculated the average results and variance. The results show that the AE-based CIG is stable with a variance of approximately 2.85% across different runs.

|  | LineWorld | | | OmniGlot | | |
|---|---|---|---|---|---|---|
|  | AE | VGG19 | PCA | AE | VGG19 | PCA |
| Reference | 64.6 ($\pm 3.57\%$) | 57.1 | 16.0 | 75.8 ($\pm 2.49\%$) | 67.2 | 28.5 |
| DFF | 33.1 ($\pm 2.23\%$) | 29.3 | 8.6 | 36.9 ($\pm 3.08\%$) | 32.2 | 11.8 |
| SCOPS | 35.6 ($\pm 3.01\%$) | 32.3 | 8.8 | 38.6 ($\pm 2.63\%$) | 35.4 | 12.0 |
| UPD | 36.3 ($\pm 2.94\%$) | 32.7 | 9.4 | 42.8 ($\pm 2.45\%$) | 37.8 | 13.2 |
| **Ours** | **58.0** ($\pm 2.79\%$) | **51.5** | **13.0** | **68.5** ($\pm 3.32\%$) | **60.6** | **20.3** |

Table 5: Experiment with different dimension reduction methods. On our method, DFF (Collins et al., 2018), SCOPS (Hung et al., 2019), and UPD (Choudhury et al., 2021).

We utilized AE as the dimension reduction technique in our experiments because PCA was not satisfactory. Since the pre-trained VGG can only be used for image data and AE yields a similar outcome to VGG, we sought a method that could provide a reasonable differentiation while being able to be applied to all types of data. We train a shared AE when comparing different methods. For each dataset, we train an AE on that dataset and then use it to calculate and compare the CIG of different methods trained on the same dataset. We run K-Means with a fixed K. For OmniGlot and ShapeNet5, we set K to 32, and for LineWorld, we select 10, which is the number of components that we consider suitable for constructing the dataset.

## J  ANALYSIS OF SHAPE SCORE

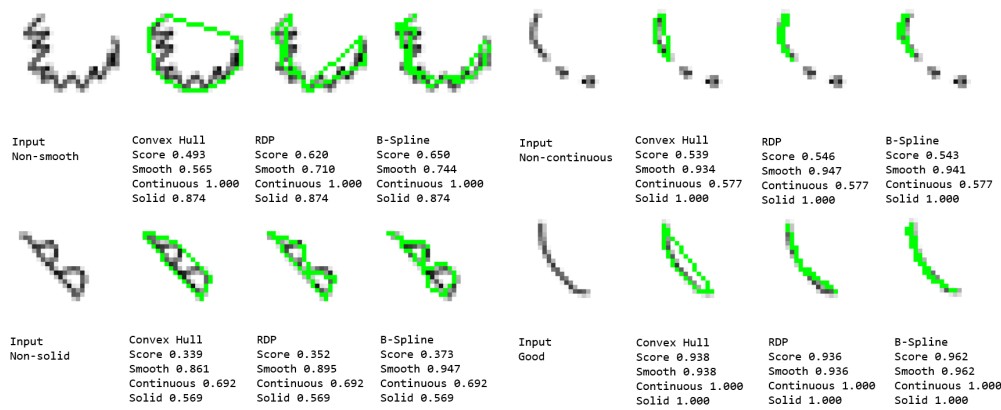

Figure 10: Visualization of the shape score under different cases and different smooth methods, the green lines are the smoothed shape.

We compare three ways to smooth the contour, minimal convex hull, Ramer–Douglas–Peucker (RDP) algorithm, and B-spline interpolation in LineWorld and OmniGlot dataset, the result is listed in Table 6, where the three methods are marked as "Hull", "RDP" and "Spline", respectively, "Reference" is the same as Table 5. "Random" is obtained with the random sampling method to calculate the CIG.

|  | LineWorld | | | OmniGlot | | |
|---|---|---|---|---|---|---|
|  | Hull | RDP | Spline | Hull | RDP | Spline |
| Reference | 89.7 | 99.5 | 97.9 | 80.0 | 86.1 | 82.9 |
| Random | 19.1 | 14.2 | 17.4 | 13.1 | 14.2 | 16.3 |
| DFF | 35.8 | 38.3 | 40.5 | 29.4 | 33.3 | 31.2 |
| SCOPS | 42.3 | 42.4 | 47.7 | 33.1 | 38.9 | 35.4 |
| UPD | 41.9 | 42.8 | 46.1 | 32.8 | 37.4 | 35.0 |
| **Ours** | **79.6** | **82.6** | **86.8** | **70.5** | **77.6** | **76.6** |

Table 6: Experiment with different contour smoothing methods. On our method, DFF (Collins et al., 2018), SCOPS (Hung et al., 2019), and UPD (Choudhury et al., 2021).

In our experiments, we apply RDP as the default smooth method, since the convex hull is too strict and prefers straight or round shapes, B-Spline and RDP give similar results, but RDP gives slightly better differentiation. We further visualize the scores under different cases and different smooth methods in Figure 10, we can see that RDP gives a smoother shape that fits the original shape better.

## K    EXAMPLES OF HUMAN EVALUATION CRITERIA

As shown on the right. In each image, different strokes are marked with different colors, the same color means one stroke. There are four options for human evaluators to choose from, as follows, depending on whether they can redraw the character with given strokes.

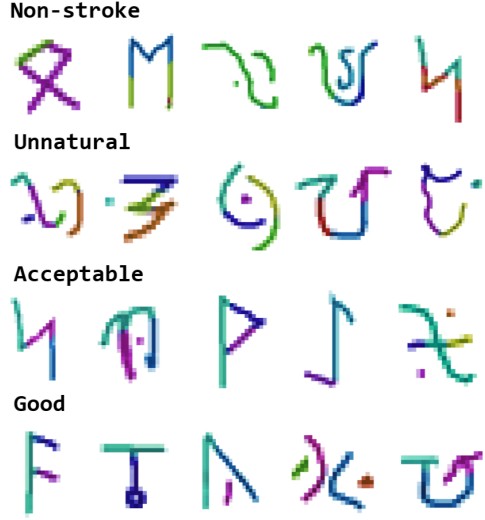

- **Non-stroke**: They are not strokes at all; it is impossible to draw the character with them.

- **Unnatural**: Can draw with these strokes, but unnatural or uncomfortable

- **Acceptable**: The strokes are not ideal enough, but not that unnatural.

- **Good**: The strokes are close to those used by humans.

Detailed instructions can be found in the Supplementary Material.

## L    LIMITATIONS AND BROADER IMPACTS

We discuss the limitations of the TDL framework in Section L.1 and its broader impacts, as well as future directions in Section L.2.

### L.1    LIMITATIONS

**Data Insufficiency.**    TDL identifies compositional and reusable predicates through multi-ary clustering on the decomposed parts. Therefore, sufficient samples are required to form clusters, and TDL will not work when not provided with enough samples. For real-life data, more samples are needed to form robust clusters considering the noise. In contrast, humans are able to find compositionality in a few-shot manner and have a high tolerance to the noise, by extrapolating the prior knowledge. This is a mystery that TDL has yet to uncover.

**Non-linear Composition.**    We assume that the input $x$ is linearly composed of parts. This simplifies the computation of the reconstruction error, as opposed to non-linear cases, where the reconstructed input needs to be obtained through a composition function that takes the set of compositions $\{x_i\}_{i=1}^{N_P}$ as input and produces the reconstructed input $\tilde{x}$. It also makes it easier to represent combinations of compositions, since they can be simply added together. However, this may limit the representation power of the model, as viewing an image as a linear combination of parts may not capture the true generation process of the input. For example, in the real world, an image is the projection of a 3D world, so the parts should be 3D, and should be reconstructed like a rendering process. Additionally, the linear assumption may not hold for other domains that also contain compositionality, such as language, audio, trajectories, etc.

**Commonsense and Reasoning.**    Some predicates can only be discovered or grounded when given a certain context. This context can be the environment, the cultural context, or common sense. Humans can also infer the missing context or deduce additional information from the input. For instance, when presented with a picture of a cat and an elephant, we can use our common sense of their sizes and the knowledge of perspective to compare their distance to us based on the size shown in the picture. This process requires a common sense of the size of the elephant and cat and reasoning ability using knowledge of perspective. However, as a representation framework, such predicates cannot be expected from TDL, as it has no common sense or a capacity to reason.

## L.2    BROADER IMPACTS AND FUTURE WORK

**General Transitional Representation.**    The TDL can be extended beyond vision, as compositionality is present in many areas, not just vision. Vision is an intuitive case for us to gain a better understanding of compositionality in high-dimensional data. The TDL looks for compositional and reusable elements or combinations as predicates from the data by treating the sample as a bag of words and the dataset as a corpus. For instance, in robotics, a trajectory is composed of reusable actions, and certain combinations of actions are known as skills. A decomposition model can be trained to suggest potential decompositions of trajectories into actions using current predicate dictionaries, and then refined through clustering the decomposed actions and combinations. By defining the decomposition models, the TDL can be used to learn the transitional representation in different domains. Another potential future direction is to learn a general cross-domain transitional representation by clustering the embeddings of multi-modal data from different decomposition models.

**Neural-Symbolic Pre-training.**    As an unsupervised representation learning framework, it is promising for TDL to scale up for large-scale pertaining. The foundation models that learn neural-symbolic transitional representations can provide better interpretability due to the embedded structural information as well as the prototype predicate dictionaries. Furthermore, the learned representation is a pre-digging of the compositional information of the data in the pre-training, which can be reused in unseen tasks, and thus theoretically perform better for downstream applications.

**Hidden Logical Rules.**    With the logical sentences of entities and relations grounded on the predicates learned by TDL, the model should be able to reason when the rules are provided. Therefore, a significant future work is learning hidden logical rules and reasoning with them in an unsupervised manner. The domain related to unsupervised learning of rules is association rule learning (Agrawal et al., 1993), which discovers rules such as $X \Rightarrow Y$ (where $X$ and $Y$ are sets of items) from the dataset. In the context of TDL, the items can be predicate prototypes. Instead of the joint probability $P(X, Y)$, a conditional probability $P(Y \, X)$ can be used to model such rules in the likelihood computation in Equation 3.

## M    QUALITATIVE COMPARISONS ON OMNIGLOT

We randomly selected 260 characters from the OmniGlot test set for comparison and qualitative analysis between our model and the baseline models. Our model, which uses a dictionary learning paradigm, can learn concepts such as lines and curves that are similar to human strokes. Each sample is colored differently; however, the colors may blend together if a pixel is associated with different parts with varying levels of confidence. The less color mixing, the higher the confidence.

## M.1 Ours

## M.2 DFF

## M.3 SCOPS

M.4   UPD

