# OpenReview forum: "Bridging Neural and Symbolic Representations with Transitional Dictionary Learning"
_ICLR.cc/2024/Conference — ICLR 2024 poster_

### Official Review · Reviewer_mxpA · 2023-10-22

**Soundness:** 3 good
**Presentation:** 3 good
**Contribution:** 3 good
**Rating:** 8
**Confidence:** 2

**Summary:**

The paper introduces Transitional Dictionary Learning – a framework for implicitly learning symbolic knowledge, such as visual parts and relations, through input reconstruction using parts and implicit relations. This is done by employing a game-theoretic diffusion model for input decomposition, leveraging dictionaries learned by the Expectation Maximization (EM) algorithm. Experimental results demonstrate the proposed approach’s efficacy through evaluation in discovering compositional patterns compared to SOTA methods, depicting human alignment with the predictions as well.

**Strengths:**

The paper provides a convincing motivation for the proposed methodology. It offers crucial insights into transitional representations, clustering information gain, and the reinforcement learning approach employed to optimize the objective. Overall, the paper exhibits a well-written supported by experimental evidence and a well-formulated mathematical framework. Figure 2, along with Section 4, elucidates the proposed approach and its crucial implementation details for the reader. I believe that this methodology holds significant promise for the research community, particularly in the midst of the surge of VLMs, where interpretable representations can not only serve as effective starting points or initializations, but also provide disentangled inputs for VLMs/LLMs to engage in high-level reasoning. The transfer learning experiments outlined in Table 2 provide strong evidence of the approach's utility beyond the confines of its training domain.

**Weaknesses:**

While the conducted experiments offer valuable insights into the effectiveness of the proposed approach, I would like to encourage the authors to extend their testing to more challenging real-world datasets. This expansion could further underscore the practical utility of the approach. Specifically, incorporating diverse categories of 3D objects from sources like ShapeNet, integrating written language datasets such as EMNIST, and including datasets featuring objects relevant to manipulation tasks would be valuable additions to the paper. Demonstrating the application of the proposed approach in contexts like robot manipulation or affordance prediction would provide tangible benefits for readers.

**Questions:**

Apart from the points mentioned in the Weaknesses section, the paper could benefit from a broader discussion of its potential applications and impact, which would be valuable for the research community. Additionally, a more detailed analysis of the computational resources and time required would be helpful for readers seeking to implement the proposed methodology.

---

> ### Author Response · Authors · 2023-11-18
> **Response to Reviewer mxpA**
>
> We sincerely thank you for your thorough review and insightful feedback. Below, we address each of your comments in detail:
>
> ## For the weaknesses:
>
> > “While the conducted experiments offer valuable insights into the effectiveness of the proposed approach, I would like to encourage the authors to extend their testing to more challenging real-world datasets. This expansion could further underscore the practical utility of the approach. Specifically, incorporating diverse categories of 3D objects from sources like ShapeNet, integrating written language datasets such as EMNIST, and including datasets featuring objects relevant to manipulation tasks would be valuable additions to the paper. Demonstrating the application of the proposed approach in contexts like robot manipulation or affordance prediction would provide tangible benefits for readers.”
>
> We are grateful for your suggestion to test our approach on more challenging real-world datasets. We recognize the potential impact and utility of expanding our experimentation to include diverse categories of 3D objects from sources like ShapeNet, as well as datasets like EMNIST and those relevant to robotic manipulation. While the current scope of our work and the limited rebuttal period constrain us from incorporating these extensions immediately, we are committed to exploring these areas in future research phases.
>
> ## For your questions:
>
> > “Apart from the points mentioned in the Weaknesses section, the paper could benefit from a broader discussion of its potential applications and impact, which would be valuable for the research community. Additionally, a more detailed analysis of the computational resources and time required would be helpful for readers seeking to implement the proposed methodology.”
>
> Your recommendation for a broader discussion on potential applications and impacts is highly appreciated. We have expanded our discussion in Appendix K.2 to encompass a broader range of applications and potential impacts of our research. This addition aims to provide the research community with a clearer vision of the future directions and the transformative potential of our methodology.
>
> We agree that a detailed analysis of computational resources and processing time would be beneficial for readers interested in implementing our methodology. We are working on it and plan to include a comprehensive efficiency analysis in future updates. This information will indeed offer valuable insights for practical applications and further research.
>
>
> ---
>
> We are confident that these responses address your concerns and clarify the directions of our research. Your suggestions have been instrumental in refining our work and guiding its future trajectory. If you have any more questions or require further clarification, please do not hesitate to reach out. Thank you once again for your valuable input and support.

---

> > ### Comment · Reviewer_mxpA · 2023-11-19
> > **Rebuttal Response**
> >
> > I am satisfied with the limitations and the broader impact statement and the future works section in the appendix.
> >
> > The considerations about the expansion of the work to real work scenarios and datasets in future work are appreciated.
> >
> > Minor comments: on page 4, top paragraph, please change: ”average” to ``average”

---

> > > ### Author Response · Authors · 2023-11-20
> > > **Thanks for your prompt response**
> > >
> > > Thank you for your prompt comment. We have carefully revised the manuscript, addressing the minor comment on page 4 and conducting a thorough review for similar typos. We remain open to any further suggestions you might have and are grateful for the opportunity to refine our work with your feedback.

---

### Official Review · Reviewer_KjjP · 2023-10-29

**Soundness:** 4 excellent
**Presentation:** 3 good
**Contribution:** 3 good
**Rating:** 8
**Confidence:** 3

**Summary:**

This paper explores unsupervised part segmentation using a neural symbolic approach.  The authors propose Transitional Dictionary Learning for symbolic feature representations for representing the feature embedding as structural information.  This is done via a set of ‘players’ estimating the visual parts which are combined together for the reconstruction and clustering losses for self-supervised learning of the features.  In addition, a game-theoretic decomposition loss prevents one player from reconstruction everything or overlapping with other players.

**Strengths:**

The paper is well-written and easy to understand.  There are good explanations for each step of the approach.  The "Transitional Representation" section does a really good job of approaching the symbolic and neural representations.


The method is topical and will be of interest to the ICLR community and the method seems to be novel for how to produce a dictionary of neuro-symbolic part segmentation.

I really like the overarching goal for self-supervised part segmentation and the method seems to attack the problem directly.  The neural symbolic approach to ML has been of interest for a while and part segmentation is a good problem to apply it towards.

**Weaknesses:**

The biggest disappointment was not doing this on real visual data rather than on LineWorld data.  This is still useful with just LineWorld but showing on realworld data would be much more impressive.

Running human evaluations requires an IRB or something similar not mentioned here.   This needs to be stated (anonymously) that you did actually go through someone to ensure the human experiments were done properly.

For the “Compositional Representation” related work, please add references to older approaches such as Bag of Words such as:
L. Fei-Fei and P. Perona, "A Bayesian hierarchical model for learning natural scene categories," 2005 IEEE Computer Society Conference on Computer Vision and Pattern Recognition (CVPR'05), San Diego, CA, USA, 2005, pp. 524-531 vol. 2, doi: 10.1109/CVPR.2005.16.

Csurka, Gabriella, Christopher Dance, Lixin Fan, Jutta Willamowski, and Cédric Bray. "Visual categorization with bags of keypoints." In Workshop on statistical learning in computer vision, ECCV, vol. 1, no. 1-22, pp. 1-2. 2004.

The citations needs to reference the actual venue such as this one should not just refer to Open Review (be wary of using automated citations):
Yann LeCun. A path towards autonomous machine intelligence version 0.9. 2, 2022-06-27. Open Review, 62, 2022.

Formular 1 -> Equation 1

**Questions:**

For Figure 3, could you compare a more conventional approach to compare against to see if this approach is causing it to be separated verse just from the data?

Have you tried this on more complex data 2D images?

Can you elaborate on exactly what the human criteria were that they were evaluating?

**Details Of Ethics Concerns:**

There was a mention of human evaluation but no mention of responsible research practices such as an IRB or something similar.

---

> ### Author Response · Authors · 2023-11-18
> **Response to Reviewer KjjP**
>
> We deeply appreciate your acknowledgement of our research topic and method, our step-by-step demonstration, and for pointing out areas for improvement. We have addressed each of your concerns as follows:
>
> ## For the weaknesses:
>
> > “The biggest disappointment was not doing this on real visual data rather than on LineWorld data. This is still useful with just LineWorld but showing on realworld data would be much more impressive.”
>
> We acknowledge the significance of applying our method to real-world data. As highlighted in our general response, this aspect is beyond the scope of our current research but is a crucial direction for future work.  For this paper, we focused on a more controlled environment to the establish the foundational aspects of our method.
>
> > “Running human evaluations requires an IRB or something similar not mentioned here. This needs to be stated (anonymously) that you did actually go through someone to ensure the human experiments were done properly.”
>
>  We appreciate your concern regarding the ethical aspects of human evaluation. To clarify, our qualitative study was conducted via Google Vertex AI. This platform ensures anonymity and ethical compliance, as we interact only with the platform, and not directly with human annotators. This setup mitigates any ethical risks related to privacy or direct human subject interaction. Guaranteed by the skilled annotators in the platform,and we annotated 3 times for each sample, the evaluation quality is also ensured.
>
> > “For the “Compositional Representation” related work, please add references to older approaches such as Bag of Words such as: L. Fei-Fei and P. Perona, "A Bayesian hierarchical model for learning natural scene categories," 2005 IEEE Computer Society Conference on Computer Vision and Pattern Recognition (CVPR'05), San Diego, CA, USA, 2005, pp. 524-531 vol. 2, doi: 10.1109/CVPR.2005.16.
>
> > Csurka, Gabriella, Christopher Dance, Lixin Fan, Jutta Willamowski, and Cédric Bray. "Visual categorization with bags of keypoints." In Workshop on statistical learning in computer vision, ECCV, vol. 1, no. 1-22, pp. 1-2. 2004.”
>
> Thank you for suggesting these seminal references. We have now incorporated citations to Fei-Fei & Perona (2005) and Csurka et al. (2004) in our related work section to enrich the historical context of our research.
>
> > “The citations needs to reference the actual venue such as this one should not just refer to Open Review (be wary of using automated citations): Yann LeCun. A path towards autonomous machine intelligence version 0.9. 2, 2022-06-27. Open Review, 62, 2022.”
>
> We have rechecked our citations to ensure accuracy and adherence to proper referencing standards. The specific citation of this Yann LeCun’s work, being available only on OpenReview, has been retained as is.
>
> > “Formula 1 -> Equation 1”
>
>    We corrected the terminology throughout the paper, replacing all instances of 'Formula' with 'Equation' for consistency and accuracy.
>
>
> ## For your questions:
>
> > ”For Figure 3, could you compare a more conventional approach to compare against to see if this approach is causing it to be separated verse just from the data?”
>
>   We have added a visualization from the UPD method in Figure 3 for comparative purposes. The caption has been updated to reflect this addition and provide clarity on the distinctiveness of our method’s results.
>
> To further answer your question, it cannot be directly caused by data, because each data point is a visual part, which cannot be straightforwardly obtained from input, and needs to be decomposed from it by the model.
>
> > “Have you tried this on more complex data 2D images?”
>
>    As mentioned, our current study does not extend to more complex or real-world 2D images, but we consider this an important area for future exploration.
>
> > “Can you elaborate on exactly what the human criteria were that they were evaluating?”
>
>   Detailed instructions and criteria for our human evaluators are provided in our supplementary materials (Data/qualitative study/Instructions.pdf) and briefly described in Appendix J. We asked annotators to assess if they could feasibly use the model’s suggested strokes to draw characters, with ratings ranging from 'Non-stroke' to 'Good' based on the naturalness and usability of the strokes:
> - Non-stroke: They are not strokes at all; it is impossible to draw the character with them.
> - Unnatural: Can draw with these strokes, but unnatural or uncomfortable
> - Acceptable: The strokes are not ideal enough, but not that unnatural.
> - Good: The strokes are close to those used by humans.
>
> The instruction file illustrates each rating with examples in detail.
>
> ---
>
> We hope that these responses thoroughly address your points and provide a clearer understanding of our research. Thank you once again for your valuable feedback and insightful questions. Should you have any further queries, please do not hesitate to contact us.

---

### Official Review · Reviewer_jEsx · 2023-10-31

**Soundness:** 2 fair
**Presentation:** 1 poor
**Contribution:** 3 good
**Rating:** 5
**Confidence:** 2

**Summary:**

The paper targets the reconstruction of an input signal $x$ (evaluated with images and point clouds) through a combination of parts in a learning framework. The solution is formulated as an unsupervised dictionary learning problem and solved through EM. The method is evaluated and compared on three datasets including 2D non-overlapping lines, 2D handwritten characters, and 3D shapes.

**Strengths:**

+ The motivation and the background of the paper are well demonstrated and insightful in Sec. 1&2. The significance of the paper is clear and the arguments are insightful.

**Weaknesses:**

The major weakness of the paper is the bad presentation of Sec. 3&4 that greatly hinders the readers from understanding the paper.
- The annotation in Sec. 3 is in quite a mess. Scalar value, vector, set, and matrix are not in consistent forms, and multiple critical variables lack clear definition/explanation:
1) what is 'a' stands for in "such as Cat(a), Tree(a), Person(a)"?
2) What is the relationship between $x_i$ and $x$? Seemingly the pieces of $x_i$ are determined by the masks and are directly combined into a whole instead of the linear addition.
3) What is the relationship between $r_i$ above Eq. 1, $R_i$ in Eq. 1, and $r_j^i$ below Eq. 1?
4) How can $theta$ be optimized in Eq. 1 if it does not appear in the two terms? The definition of the decoder g(·) is not consistent. Does it take $theta$ as a condition or not? Seemingly Eq. 2 is the appropriate form.
5) The definitions of two crical terms $E_{\tilde{D}}$ and $d_S$ are unclear.
6) How is the dictionary $\tilde{D}$ obtained given the argument "As we have meaningful $\tilde{D}$"?
7) It seems that the only variable to be optimized is the hidden dictionary $\theta$. What about the models of $f(x;\theta)$, $\hat{g}(r_i;\theta)$, $g_{\theta}(R^i)$, and $g_{\tilde{D}(R)}$?

- The illustration of Fig. 2 does not clearly demonstrate the formulation in Sec. 3 and the solution in Sec. 4:
1) $f, R, r_i, g, x_i, m_i$ are not clearly labeled in the figure.
2)Where is the $N_P$ copies of the model in the figure?
3) What does each patch stands for and what are the relation between the patches and the aforementioned terms in Sec. 3?
4) Why is there a "GT loss" in an unsupervised learning pipeline?
5) Where is the "Decomposition Loss" mentioned in Fig. 2?

**Questions:**

Though Sec. 1&2 are well-demonstrated with clear motivations, the unsatisfied presentation of Sec. 3&4 makes the formulation and solution hard to follow. The authors are also encouraged to provide qualitative comparison results on Line World and ShapeNet5.

---

> ### Author Response · Authors · 2023-11-18
> **Response to Reviewer jEsx (Part 1/2)**
>
> We greatly appreciate your acknowledgment of our research problem demonstrated in Sections 1 and 2, and the significance of our paper. We are also highly grateful for your comprehensive review and constructive, concrete feedback on Sections 3 and 4. We have revised our manuscript significantly, addressing each of your points for clarity and coherence:
>
>
> ## For the weaknesses:
>
> > “The major weakness of the paper is the bad presentation of Sec. 3&4 that greatly hinders the readers from understanding the paper.”
>
> We have substantially revised Sections 3 and 4 for better clarity and coherence, with changes highlighted in blue in the revised manuscript. These revisions are intended to address the issues you raised and improve overall readability.
>
> > “The annotation in Sec. 3 is in quite a mess. Scalar value, vector, set, and matrix are not in consistent forms, and multiple critical variables lack clear definition/explanation:”
>
> We meticulously reviewed and standardized our notation, adopting a consistent sub/super-script convention detailed at the start of Section 3: ***“Superscript $·^i$ denotes $i$-arity, superscript $·^{(i)}$ with brackets indicates the $i$-th sample in a dataset, and subscript $·_i$ refers to the $i$-th visual part in a sample.”*** This clarification aims to resolve any ambiguity in our notation.
>
>
>
> > “what is 'a' stands for in "such as Cat(a), Tree(a), Person(a)"?”
>
> We clarified that ‘a’ represents a logical variable, revising examples to use $x_i$ (e.g., Cat($x_i$)) directly for greater clarity.
>
> > “What is the relationship between $x_i$ and $x$? Seemingly the pieces of  are determined by the masks and are directly combined into a whole instead of the linear addition.”
>
> $x_i$ is a visual part of image $x$, it is true that under our linear assumption, $x=\sum_{i=1}^{N_P} x_i=\sum_{i=1}^{N_P} m_ix$, the parts can be decided by masks, that is the reason why we introduce the game theoretic mechanisms to avoid collapsed solutions, i.e. directly output an all-1 mask.
>
> We rewrote paragraph 3 in Section 3.1 to introduce the linear assumption in detail first, followed by a discussion on symbol grounding, in order to reduce such ambiguity.
>
> > “What is the relationship between $r_i$ above Eq. 1, $R_i$ in Eq. 1, and $r^i_j$ below Eq. 1?”
>
> We solved this kind of ambiguity by the sub/superscript convention discussed above. $R^{(i)}$ is the transitional representation or neural logical variables (each variable is an visual part) for $i$-th sample in the dataset, $r_i$ is the representation for the i-th part in $R$, $r^{(i)}_j$ is the representation for the $j$-th part in $i$-th sample.
>
> > “How can $theta$ be optimized in Eq. 1 if it does not appear in the two terms? The definition of the decoder $g(·)$ is not consistent. Does it take $theta$ as a condition or not? Seemingly Eq. 2 is the appropriate form.”
>
> Yes, the theta (dictionary) should be used in both $f$ and $g$, while $f$ and $g$ (the diffusion model) should also have their own parameters to be optimized.
>
> We included theta (dictionary) in both the $f$ and $g$ and provided additional clarification on how $R$ is derived from $f$ in Equation (1).
>
> > “The definitions of two crical terms $E_{\tilde{D}}$ and $d_S$ are unclear.”
>
> We expanded the explanation of the expectation term and the ideal metric distance in the revised paper by rewriting paragraph 5 in Section 3.1, we introduced details and examples to clarify these critical concepts.
>
> > “How is the dictionary $\tilde{D}$ obtained given the argument "As we have meaningful $\tilde{D}$"?”
>
> We fixed this ambiguous sentence by rewriting Section 3.1 paragraph 5, it is not we have meaningful dictionaries, but we only need to consider meaningful dictionaries. We also updated the corresponding sentence to ***“As we only need to consider meaningful dictionaries…”***
>
>
> > “It seems that the only variable to be optimized is the hidden dictionary $\theta$. What about the models of $f(x;\theta)$, $\hat{g}(r_i;\theta)$, $g_\theta(R^i)$, and $g_{\tilde{D}}(R)$”
>
> We clarified that the parameters of the decomposition model ($f$ and $g$) are also optimized, alongside the dictionary, to resolve any confusion and added this sentence: ***“we omit the parameters of decomposition model f and g that also need to be optimized in this target for simplicity.”***.

---

> ### Author Response · Authors · 2023-11-18
> **Response to Reviewer jEsx (Part 2/2)**
>
> > “The illustration of Fig. 2 does not clearly demonstrate the formulation in Sec. 3 and the solution in Sec. 4:”
>
>  Figure 2 has been updated to better align with the formulation in Section 3, including annotated mathematical notations and more details about the player models.
>
> > 1. “ are not clearly labeled in the figure. 2)Where is the  copies of the model in the figure?”
>
> We added labels to the figure to clarify them.
>
> > 2. “What does each patch stands for and what are the relation between the patches and the aforementioned terms in Sec. 3?”
>
> We explained the patches as visual parts corresponding to the set of visual parts {$x_i$}$_{i=1}^{N_P}$.
>
> > 3. “Why is there a "GT loss" in an unsupervised learning pipeline?”
>
> GT loss is to avoid collapsed outputs as discussed above. It is unsupervised as it does not rely on any label. It introduced additional inductive biases to remove collapsed solutions from its search space.
>
> > 4. “Where is the "Decomposition Loss" mentioned in Fig. 2?”
>
> Decomposition loss is written at the end of Section 4.1, we bolden it, we also renamed the Reconstruction loss to reconstruction error as it is a part of decomposition loss to avoid such confusion.
>
>
> ## For your questions:
>
>
> > “Though Sec. 1&2 are well-demonstrated with clear motivations, the unsatisfied presentation of Sec. 3&4 makes the formulation and solution hard to follow. The authors are also encouraged to provide qualitative comparison results on Line World and ShapeNet5.”
>
> The concerns about Sections 3 and 4 are addressed above. We acknowledge the importance of providing qualitative comparisons on Line World and ShapeNet5. We will include these additional insights in the final version of our paper.
>
> ---
>
> We have made these revisions to directly address each of your points to ensure that our paper is as clear and informative as possible. Thank you again for your invaluable input. We hope our updates can address your concerns and help you re-evaluate our paper. We are open to further discussions and feel free to ask us any questions you have.

---

### Official Review · Reviewer_6UkK · 2023-11-01

**Soundness:** 2 fair
**Presentation:** 1 poor
**Contribution:** 2 fair
**Rating:** 5
**Confidence:** 4

**Summary:**

This paper looks at a way to merge symbolic and DNN representations. The authors propose a transitional representation that contains high-fidelity details of the input and and also provides structural information about the semantics of the input. An Expectation-Maximization loop is used to optimize the parameters, where the Expectation step is used to optimize the hidden dictionary of parts, and maximize the overall likelihood of the dataset. To control the arity, techniques such as online clustering and random sampling are used. The authors conduct unsupervised segmentation on three abstract compositional visual object datasets and show superior accuracy compared to unsupervised clustering baselines.

**Strengths:**

Neuro-symbolic reasoning is a timely topic for research, and joint optimization of reconstruction and predicate logic appears to be an interesting idea. The method utilizes a dictionary of entities, and 1-any and 2-ary predicates as a neck to train the semantic distance during reconstruction. The works incorporates several interesting ideas such as Expectation Maximization, game-theoretic loss function and online prototype clustering to make the system work.

**Weaknesses:**

- The paper is a hard to read and the language is confusing. Technical concepts such as "hidden dictionaries of symbolic knowledge" are introduced early on without much explanation.
- Experiments are limited to tiny, mostly binary datasets such as "ShapeNet5", which is basically a subset of 5 categories from the ShapeNet dataset. It is not clear if the methods would generalize to noisy real-world data, such as training using noisy, incomplete instances where the parts are not all visible.
- Although the paper provides a reasonably well-curated list of neuro-symbolic approaches, the evaluations do not compare against any of the recent approaches. Instead the comparison is against clustering baselines.
- The paper reads as a mishmash of several different ideas that are used together, but not integrated coherently. Therefore having a ablation studies to show the value of each module would be crucial. However, the evaluations do not provide a clear understanding of the contribution of each component to the overall methodology.
- Ultimately, the task of reconstructing and explaining shapes simultaneously might be quite ambiguous as depicted in figure 4, and might not generalize to natural datasets, These aspects are not addressed in the paper.

**Questions:**

1. Are the predicates shared among different classes? Do predicates always correspond to semantic attributes? It would help to visualize the learnt 1-any and the 2-ary predicates.
How does the method compare to other neuro-symbolic baselines? The current set of baselines are essentially unsupervised clustering methods.
2.
2. Please provide a clear set of ablation studies which show the benefits drawn from each component. How can the system be simplified without affecting the overall accuracy.
3. It would be good to have a limitations section that discusses when this method wouldn't work. How do predicates such as left_of and larger (examples from the paper) operate in case of multi-view settings, where these terms become ambiguous.

---

> ### Author Response · Authors · 2023-11-18
> **Response to Reviewer 6UkK (Part 1/2)**
>
> We sincerely appreciate your acknowledgment of our research topic and our idea, and your constructive feedback and comprehensive review. Our responses to your concerns are as follows:
>
> ## For the weaknesses:
>
> > “The paper is a hard to read and the language is confusing. Technical concepts such as "hidden dictionaries of symbolic knowledge" are introduced early on without much explanation.”
>
> As mentioned in the response to all reviewers, we have updated our technical writing in the revised version. Specifically, in the introduction, we updated the corresponding sentence to:
> ***“TDL uses an Expectation Maximization (EM) algorithm to iteratively update dictionaries that store hidden representations of symbolic knowledge, through an online prototype clustering on the visual parts decomposed from the inputs by a novel game-theoretic diffusion model using the current dictionaries”***
>
> It uses only terms like “hidden representations” which are widely accepted in ICLR community. It also directly expresses the relation of EM algorithm, game-theoretic decomposition, and the prototype clusterin to avoid confusion. We also checked the entie paper to avoid using new terms without explanation.
>
> > “Experiments are limited to tiny, mostly binary datasets such as "ShapeNet5", which is basically a subset of 5 categories from the ShapeNet dataset. It is not clear if the methods would generalize to noisy real-world data, such as training using noisy, incomplete instances where the parts are not all visible.”
>
> Our experimental setup is delicately designed to exclude visual feature confoundings. This approach substantiates our main claim that TDL can learn compositional and reusable visual predicates from high-dimensional input. Extension to noisy real-world data is recognized as an important future direction that is out of the main scope of this paper.
>
> > “Although the paper provides a reasonably well-curated list of neuro-symbolic approaches, the evaluations do not compare against any of the recent approaches. Instead the comparison is against clustering baselines.”
>
> The neuro-symbolic approach is a broad concept, it can cover almost all kinds of tasks while baseline picking is task-specific. In our task, unsupervised learning of concepts and relations (predicates) from visual data, the unsupervised part segmentation methods are the most relevant. To the best of our knowledge, we do not see a suitable, directly comparable neural symbolic baseline, especially for this problem.
>
> > “The paper reads as a mishmash of several different ideas that are used together, but not integrated coherently. Therefore having a ablation studies to show the value of each module would be crucial. However, the evaluations do not provide a clear understanding of the contribution of each component to the overall methodology.”
>
> We improved our writing to make the structure of our method clear. Our method is basically an EM algorithm implemented by two components, clustering and a decomposition model (i.e. the game-theoretic diffusion model). We provided the ablation study in Appendix E, “Base” vs ”+Cluster”, the model with and without clustering, we further provided an ablation study on the commonly used optimizations for the decomposition model.
>
> Inside the decomposition model, its loss terms are all necessary to guarantee the right function (right convergence) of the model, the reconstruction error is the basis, and the other GT loss terms are designed to avoid collapsed outputs which are further discussed in Appendix D, SMLD loss is to train the scoring based diffusion model we used. They cannot be further decomposed into an ablation study.
>
> > “Ultimately, the task of reconstructing and explaining shapes simultaneously might be quite ambiguous as depicted in figure 4, and might not generalize to natural datasets, These aspects are not addressed in the paper.”
>
> We modified the caption of Figure 4 from:
> ***“Comparison between our method and baselines on the OmniGlot test set. Our method can learn interpretable strokes compared to the baselines that failed to give effective strokes.”***
>  to:
> ***“Examples from OmniGlot test set. Our method generates multiple interpretable strokes to reconstruct the input hand-written characters. As a comparison, the baseline methods segment the input into colored parts that are not valid strokes revealing a failure in learning compositionality.”***
>
> It introduced more details about the difference between our method and baselines, and why our method can be regarded as successful in learning reusable predicates.
>
> As discussed earlier, It is natural to extend the basic TDL proposed to natural data, it firstly requires considering the noise as discussed, and secondly the additional information, including the visual features like edges, shadows, and textures, which are normally related to the use of pre-trained backbones.

---

> ### Author Response · Authors · 2023-11-18
> **Response to Reviewer 6UkK (Part 2/2)**
>
> ## For your questions:
>
> > 1.1 Are the predicates shared among different classes?
>
>
> Yes, as an unsupervised learning method, the model is unaware of “classes”.  In our experiments, the learned predicates correspond to visual parts and combinations (e.g., armrest, leg), applicable across different object types (e.g. table, chair), It is also demonstrated in our ShapeGlot transfer learning to unseen classes experiment.
>
>
> > 1.2 Do predicates always correspond to semantic attributes? It would help to visualize the learnt 1-any and the 2-ary predicates.
>
> We rewrote Section 3.1 paragraph 5 to reduce the confusion related to this question. To answer your question, the correspondence of predicates to semantic attributes is not guaranteed, as it is subjective like there exist different fonts to write the same character with the same meaning. The learned transitional representation (i.e. predicates) is an "average" of those fonts that have the minimal possible distance to all of them by approximating their commonality, the reusable and compositional, as demonstrated in the objective target in Section 3.2. And each font is a "concrete meaningful dictionary" mentioned in Section 3.2. Such an "average" dictionary is not necessarily highly meaningful but should be close enough as we assume the meaningful ones are compositional and reusable.
>
> We agree that visualizing the latent predicates is helpful, right now we do not have a decoder that can directly decode the predicate as an image, we are working on it and try to update it in the final version.
>
> > 1.3 How does the method compare to other neuro-symbolic baselines? The current set of baselines are essentially unsupervised clustering methods.
>
> As discussed above in Weaknesses 3, to the best of our knowledge, we cannot find a neuro-symbolic baseline for our specific task.
>
> > 2. Please provide a clear set of ablation studies which show the benefits drawn from each component. How can the system be simplified without affecting the overall accuracy.
>
> Addressed in Weaknesses 4 above. Further simplification might involve merging loss terms or integrating the game theory mechanism into a multi-body diffusion process, which we consider beyond this paper’s scope.
>
> > 3. It would be good to have a limitations section that discusses when this method wouldn't work. How do predicates such as left_of and larger (examples from the paper) operate in case of multi-view settings, where these terms become ambiguous.
>
> We've added a new section in Appendix K.1 discussing the limitations. Particularly, the model's output may be ambiguous in scenarios where the problem itself is ambiguous to humans. For example, A is on the left of B in image 1, while A is on the right in image 2. Decide whether A is on the left or right will be an undecidable problem, the model will also give an ambiguous output. Similar to Figure 8, bottom left, when the model is given an invalid combination of parts, it cannot decide a clear relation.
>
> Relations like "larger" are different as it is related to reasoning (i.e. using knowledge/rule of perspective) which is not involved in TDL as a representation method. We added the discussion of the unsupervised rule learning in Appendix K.2 under the view of TDL.
>
> ---
>
> We hope our replies can address your concerns and help you further evaluate our paper, and we are happy to solve any additional questions you may have and further improve our paper. Thank you again for your invaluable feedback.

---

### Author Response · Authors · 2023-11-18
**Response to all Reviewers**

We express our gratitude to all the reviewers for their valuable feedback. We reply to common concerns in this general response. We have also responded to each reviewer individually to address any comments/concerns.

## Presentation updates

We thank reviewers jEsx, KjjP, and mxpA for recognizing our clear presentation of the problem and background. In response to feedback from reviewers 6UkK and jEsx, we have made substantial revisions to enhance the clarity of method writing (highlighted in blue in the revised version), we summary those edits here:

1. **Introduction (Paragraph 4):** We've replaced terms like “hidden dictionaries of symbolic knowledge” with “dictionaries storing hidden representations of symbolic knowledge” using widely accepted terms. we also updated this paragraph to summarize the TDL framework and clarify the connection between introduced concepts and applied methods.

2. **Related Work - Compositional Representation:** Added references to classic approaches for a more comprehensive perspective.

3. **Section 3 Begining:** Introduced a consistent sub/super-script convention and uniformly updated notations across the paper.

4. **Section 3.1 (Paragraph 3):** Detailed the linear assumption, move it before the discussion on symbolic representation construction.

5. **Section 3.1 (Paragraph 5):** Provided clear explanations and examples to illustrate concepts like expected semantic distances and concrete dictionaries. Added a short description of how the learned dictionary of transitional representation can be applied in practice.

6. **Equation (1):** Explained $R$ is generated by $f$. Clarified model parameters are omitted. Added details on ideal semantic distance and the expectation term in the following interpretation paragraph.

7. **Section 3.2:** Added more details to explain alternative target, summary about how the proposed EM algorithm works, and likelihood computation. Added a sentence that leads readers to the new discussion appendix.

8. **Section 4:** Simplified U-Net description, and clarified the visual parts are generated by multiple player models.

9. **Figure 2 Update:** Added mathematical notations that aligned with text, added explanatory blocks of player model, and renamed “reconstruction loss” to “reconstruction error.” and made it shallow color.

10. **Sections 4.1 & 4.2:** Clarified terminologies. Change m in the memory bank to mu to avoid confusion with the mask.

11. **Figure 3:** Added the visualization of the latent space from the baseline UPD, and updated the caption, and description in Section 4.2.

12. **Figure 4 Caption Update:** Clarified distinctions between our method and baselines.

13. **Section 5.1:** Emphasized rationale behind selecting abstract visual object tasks that exclude confounding.

14. **Appendix K:** A new section discusses limitations, broader impacts, and future work.



## Regarding the Lack of Real-world Dataset Concerns

We delicately designed our experiments to avoid the confoundings from common visual features such as edges and textures. This ensures that the model learns reusable entities and relations (predicates)  instead of a simple edge detector from high-dimensional inputs in an unsupervised manner when it performs well in these tasks. This is a significant and previously unaddressed challenge and it is the major claim of this paper. Our experimental results sufficiently supported our contribution to this problem.

To respond to the concerns from reviewers 6UkK and KjjP about real-world datasets: we recognize the potential benefits but consider them to be an extension beyond the current scope. The adaptation to real-world datasets would involve incorporating additional error terms in Equation (1) and adjustments for denoising and noise reconstruction in the encoder and decoder, respectively. And a clustering on the visual parts after denoising. While these are crucial and logical extensions that can naturally fit into the TDL framework, we opted to focus on a more concentrated scope within the limited rebuttal period, leaving these aspects for future work.

---

### Meta-Review · Area_Chair_NHST · 2023-12-09

**Metareview:**

The paper introduces Transitional Dictionary Learning (TDL), a framework that learns symbolic knowledge by reconstructing input as a combination of parts with implicit relations, showing superior performance in compositional pattern discovery over traditional methods, with results validated by human evaluation and novel metrics. Two reviewers agreed to accept the paper and two reviewers recommended rejection. While the reviewers agree the topic and approach are interesting and novel, there are concerns about the experiments are relatively toy. After carefully reading the paper and the authors' answers, the AC find it still can be an interesting finding for the ICLR community and recommend acceptance.

**Justification For Why Not Higher Score:**

The experiments are still relatively toy.

**Justification For Why Not Lower Score:**

The approach is sufficiently interesting.

---

### Decision · Program_Chairs · 2024-01-16

Accept (poster)